# Brain size and neuron numbers drive differences in yawn duration across mammals and birds

Jorg J. M. Massen [1,7✉], Margarita Hartlieb[2,7], Jordan S. Martin[3,7], Elisabeth B. Leitgeb[2], Jasmin Hockl[2], Martin Kocourek[4], Seweryn Olkowicz [4], Yicheng Zhang [4], Christin Osadnik[5], Jorrit W. Verkleij[1], Thomas Bugnyar [2], Pavel Němec [4] & Andrew C. Gallup [6✉]

Recent studies indicate that yawning evolved as a brain cooling mechanism. Given that larger brains have greater thermolytic needs and brain temperature is determined in part by heat production from neuronal activity, it was hypothesized that animals with larger brains and more neurons would yawn longer to produce comparable cooling effects. To test this, we performed the largest study on yawning ever conducted, analyzing 1291 yawns from 101 species (55 mammals; 46 birds). Phylogenetically controlled analyses revealed robust positive correlations between yawn duration and (1) brain mass, (2) total neuron number, and (3) cortical/pallial neuron number in both mammals and birds, which cannot be attributed solely to allometric scaling rules. These relationships were similar across clades, though mammals exhibited considerably longer yawns than birds of comparable brain and body mass. These findings provide further evidence suggesting that yawning is a thermo-regulatory adaptation that has been conserved across amniote evolution.

---

[1] Animal Behaviour and Cognition, Department of Biology, Utrecht University, Utrecht, The Netherlands. [2] Department of Behavioral & Cognitive Biology, University of Vienna, Vienna, Austria. [3] Human Ecology Group, Institute of Evolutionary Medicine, University of Zurich, Zurich, Switzerland. [4] Department of Zoology, Charles University, Prague, Czech Republic. [5] Department of General Zoology, University of Duisburg-Essen, Essen, Germany. [6] Psychology Program, Department of Social and Behavioral Sciences, SUNY Polytechnic Institute, Utica, NY, USA. [7] These authors contributed equally: Jorg J. M. Massen, Margarita Hartlieb, Jordan S. Martin. ✉email: jorgmassen@gmail.com; a.c.gallup@gmail.com

Yawning is a stereotyped action pattern characterized by an involuntary and powerful gaping of the jaw with deep inspiration, followed by a temporary period of peak muscle contraction and a passive closure of the jaw with shorter expiration[1]. Various lines of evidence suggest that yawning is an adaptation, including comparative analyses across diverse vertebrates[2], the early emergence of yawns during embryological development[3], psychological research indicating hedonic properties of yawning[4], and clinical findings on the potential costs of this response (i.e., subluxation or locking of the lower jaw[5]).

While dozens of hypotheses have been put forth to explain the biological significance of yawning[6], very few have garnered any empirical support[7,8]. This includes the still popular hypothesis that yawns function to increase or equilibrate oxygen in the blood, which was falsified over 30 years ago. In particular, an elegant set of studies demonstrated that altered levels of oxygen and carbon-dioxide do not alter yawning, and physical exercise sufficient to double breathing rates has no effect on yawning[9]. Thus, it has been concluded that yawning and breathing are triggered by distinct internal states and controlled by different mechanisms.

To date, the brain cooling hypothesis is the most strongly supported explanation for why we yawn. This hypothesis proposes that the extended muscular contractions and deep inhalation during the act of yawning function to flush hyperthermic blood away from the skull, while simultaneously introducing cooler arterial supply through convective heat transfer and evaporative heat loss[10–12]. In support of this hypothesis, the natural expression of yawning coincides with predicted changes in brain temperature[13], oral temperature[14], and surface skull temperature[15] before and after the motor action pattern. That is, yawns are triggered during rises in temperature, and are followed by corresponding decreases in temperature. Yawn frequency is also reliably inhibited by methods of behavioral brain cooling in humans[10,16], including the cooling of the carotid arterial blood supply, which is sufficient to decrease temperature at the superiomedial orbital area (i.e., the brain temperature tunnel)[16]. Moreover, as predicted by behavioral thermoregulatory models, changes in ambient temperature directly alter the rate of yawning across diverse species[17–20].

The brain cooling hypothesis further predicts that, in order to achieve the same functional outcome, the duration of yawns should be tied to brain size and neuron numbers across animals. Larger brains have greater thermolytic needs, i.e., as brains increase in size, this necessitates greater physiological changes to dissipate heat and maintain thermal homeostasis[21]. In addition, metabolic heat production from neural activity is one of the principal factors that determines the temperature of the brain[22–25]. Therefore, animals with larger and more complex brains should require longer and more powerful yawns to produce comparable cooling effects. In support of this prediction, an initial study on 24 mammalian species found strong correlations between average yawn duration and the brain mass and cortical neuron number of these species[26]. A subsequent study found a similar relationship between brain volume and yawn duration within a single family of mammal species; i.e., Felidae[27], and more recently a strong positive relationship between brain mass and yawn duration has even been found within a single species; i.e. domesticated dog breeds[28]. All three studies suggest that yawning evolved to serve an important neurophysiologic function. However, these studies suffered from a relatively small sample size, failed to control for phylogenetic history, and were restricted to mammals only, hampering any generalization of these results across other taxa.

Birds, as the only other endothermic class of animals, may equally benefit from thermoregulatory mechanisms to maintain brain homeostasis. Most bird species have a morphological adaptation, the rete mirabile ophthalmicum, which makes energy transfer between the brain and the bloodstream more efficient and allows for selective brain cooling[29,30]. Nevertheless, this system is still reliant on the delivery of cooler blood in the vessels to the rete mirabile ophthalmicum and both oral and nasal cavities have been established as sites of thermal exchange between the blood and the environment[31,32]. Consequently, birds also benefit from behavioral adaptations that increase thermal exchange, as is hypothesized for yawning. Consistent with this view, yawning in budgerigars (Melopsittacus undulatus) is altered by changes in ambient temperature similar to mammals[33,34], and thermal imaging has revealed significant decreases in facial temperature following yawns in this species[35]. Yet, yawning in birds remains very much understudied and especially large-scale comparative analyses on bird yawns are lacking.

## Results

**Yawn duration, brain mass, and neuron numbers**. A total of 1291 yawns (622 mammal and 669 bird) from 697 different individuals (426 mammals and 271 birds) across 101 different species (55 mammalian and 46 avian) were analyzed for their duration according to the definition of yawning put forward by Barbizet[1] (see Fig. 1). Yawns were checked for validity and reliability by four different researchers (see "Methods" section). Body mass and brain mass of all species were extracted from the literature. Similarly, total number of brain neurons and number of cortical/pallial neurons for mammalian and avian species were extracted from the literature, while neuron counts for an additional ten avian species were newly added here (see "Methods" section for specifics). All neuron number estimates used in this study are based on the isotropic fractionator[36]. Since the total brain neuron numbers and cortex/pallium neuron numbers were not yet known for all 101 species in our sample, these analyses were performed on a restricted sample (total neuron number, $n = 18$ and 33 species, and cortex/pallium neuron numbers, $n = 19$ and 33 species for mammals and birds, respectively). Bayesian multilevel phylogenetic models were used to investigate the associations between yawn duration and brain measures while accounting for species' phylogenetic history, as well as to control for any unobserved heterogeneity between species not explained by phylogeny (see Fig. 2; see "Methods" section for specifics).

Model comparison did not support the inclusion of random slopes across either mammalian ($\triangle$WAIC $= -1.67$ [1.80]) or avian orders ($\triangle$WAIC $= -0.60$ [1.70]), indicating that the observed effects of brain measures on yawn duration were consistent across taxa. Within mammals (Fig. 3), a large positive association was found between yawn duration and standardized brain mass ($\beta = 0.35$ [0.06], 90% CI [0.25, 0.45], $p_+ = 1.00$, $d = 1.37$ [0.22]), as well as between yawn duration and the total neuron count ($\beta = 0.35$ [0.10], 90% CI [0.18, 0.51], $p_+ = 1.00$, $d = 1.25$ [0.36]) and cortical neuron count ($\beta = 0.36$ [0.12], 90% CI [0.16, 0.55], $p_+ = 0.99$, $d = 1.28$ [0.41]).

Among avian taxa, a large positive association was found between yawn duration and standardized brain mass ($\beta = 0.20$ [0.05], 90% CI [0.13, 0.28], $p_+ = 1.00$, $d = 0.71$ [0.16]), with moderately sized associations observed between yawn duration and the total neuron count ($\beta = 0.18$ [0.05], 90% CI [0.11, 0.26], $p_+ = 1.00$, $d = 0.61$ [0.16]) and pallium neuron count ($\beta = 0.20$ [0.05], 90% CI [0.12, 0.27], $p_+ = 1.00$, $d = 0.66$ [0.16]) (Fig. 3).

**Brain/body allometry and correcting for body mass**. The aim of this study was to examine how absolute brain mass and neuron numbers drive differences in yawn duration across mammals and birds, yet models that do not control for body mass fail to rule out

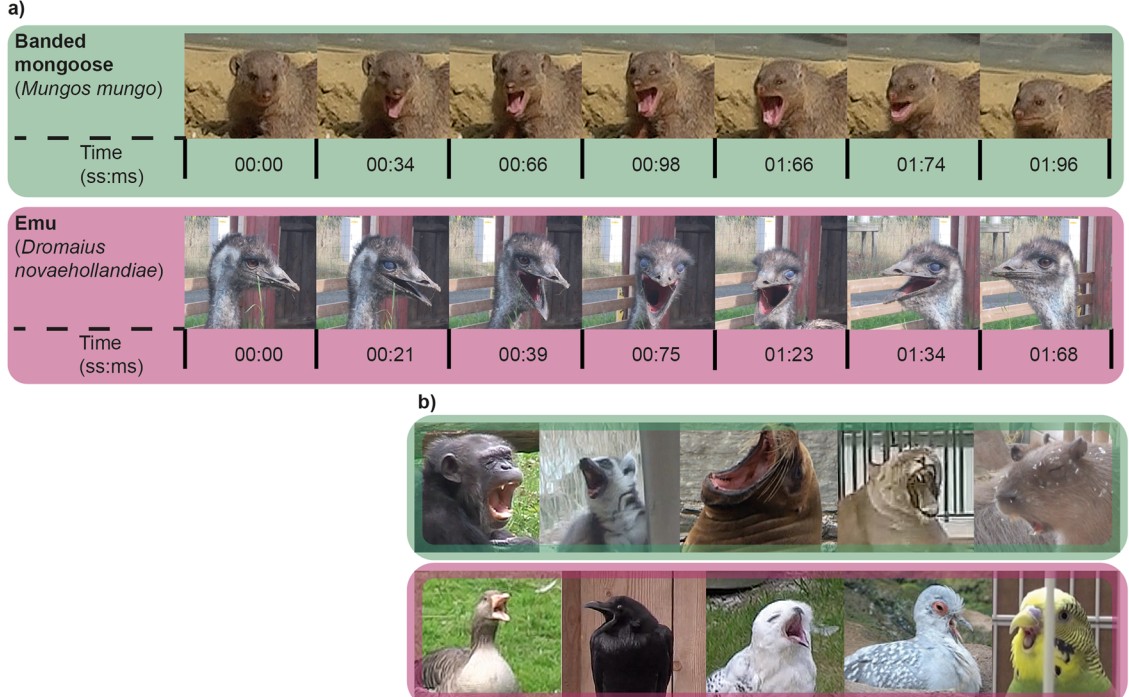

**Fig. 1 Examples of yawns from mammals and birds. a** Video sequence of a yawn from both a mammal (Banded mongoose) and a bird (Emu), which demonstrate the fixed action pattern as described by Barbizet[1], including the twitching of the eyes in the banded mongoose and the closing of the nictitating membrane in the emu. **b** Images of several species in our sample mid-yawn. Above: Mammals, from left to right: Chimpanzee (*Pan troglodytes*), Ring-tailed lemur (*Lemur catta*), Southern sea lion (*Otaria flavescens*), Lion (*Panthera leo*), and Capybara (*Hydrochoerus hydrochaeris*); below: Birds, from left to right: Greylag goose (*Anser anser*), Common raven (*Corvus corax*), Snowy owl (*Nyctea scandiaca*), Diamond dove (*Geopelia cuneata*), and Budgerigar (*Melopsittacus undulatus*).

byproduct explanations due to allometric brain/body scaling. To avoid biases due to multicollinearity between body mass and the respective brain measures, we calculated residual brain mass measures from a phylogenetically controlled linear regression of the respective brain measure on body mass[37] and used these residuals as predictors in subsequent analyses. If, however, the risk of such a bias due to multicollinearity was acceptably low, we rather included body size as an additional predictor in the original models (see "Methods" section for specifics).

Similar to the models for absolute brain measures, model comparison did not support the inclusion of random slopes across either mammalian ($\triangle$WAIC $= -0.51$ [1.90]) or avian orders ($\triangle$WAIC $= -3.16$ [3.12]), suggesting that the observed effects of the body size adjusted brain measures on yawn duration were also consistent across taxa. Within mammals, after controlling for body size, clear positive associations remained between yawn duration and mammalian brain size ($\beta = 0.11$ [0.07], 90% CI [0.00, 0.24], $p_+ = 0.95$, $d = 0.45$ [0.28]), with similarly sized but more statistically uncertain associations observed between yawn duration and the total mammalian neuron count ($\beta = 0.12$ [0.14], 90% CI [$-0.11$, 0.35], $p_+ = 0.80$, $d = 0.42$ [0.49]) and cortical neuron count ($\beta = 0.13$ [0.16], 90% CI [$-0.16$, 0.39], $p_+ = 0.78$, $d = 0.46$ [0.58]) (Fig. 4). The near equivalence of these effect sizes is expected, as the mammalian neuron counts exhibited very high phylogenetic correlations with total brain size (total neuron count: $r_{phylo} = 0.98$; cortical neuron count: $r_{phylo} = 0.93$), suggesting that these measures are providing similar information about brain evolution. Moreover, given the much smaller sample size for the neuron count measures ($N_{species} = 18$–$19$) as compared to total brain size ($N_{species} = 55$), which resulted in very low power for detecting small effect sizes

(power $= 0.08$ for $\beta = 0.10$; see Supplementary Note 1), it is expected that the neuron count effects would exhibit greater statistical uncertainty than total brain size.

Among avian taxa, after adjusting for body size, moderately sized positive associations remained between yawn duration and brain size ($\beta = 0.17$ [0.10], 90% CI [0.00, 0.33], $p_+ = 1.00$, $d = 0.58$ [0.35]), as well as between yawn duration and total neuron count ($\beta = 0.20$ [0.08], 90% CI [0.06, 0.34], $p_+ = 0.99$, $d = 0.67$ [0.28]) and pallium neuron count ($\beta = 0.18$ [0.08], 90% CI [0.05, 0.30], $p_+ = 0.99$, $d = 0.61$ [0.26]) (Fig. 4). As in mammals, the phylogenetic correlations between the neuronal count and brain size measures were very high across birds (total neuron count: $r_{phylo} = 0.94$; pallium neuron count: $r_{phylo} = 0.91$).

To further examine the robustness of these findings, we also conducted analyses, in which we first used Gaussian phylogenetic regressions to partial out the effects of body size on *both* the brain measures as well as yawn duration prior to estimating their association. This approach relied on the simplifying assumption that yawn duration residuals are normally distributed, and we therefore used classical linear PGLS models for assessing the association among these residual values. Consistent with the findings using more robust Bayesian methods, significant associations between residual yawn duration and all residual brain measures were observed across mammals and birds (see Supplementary Note 2).

Comparisons of the different models did not render any clear differences between the explanatory strength of models with brain size, total neuron count, and cortical/pallial neuron count in both mammals and birds (see Supplementary Note 3). It is important to emphasize, however, that these findings should be interpreted with caution. These results are expected given the small sample

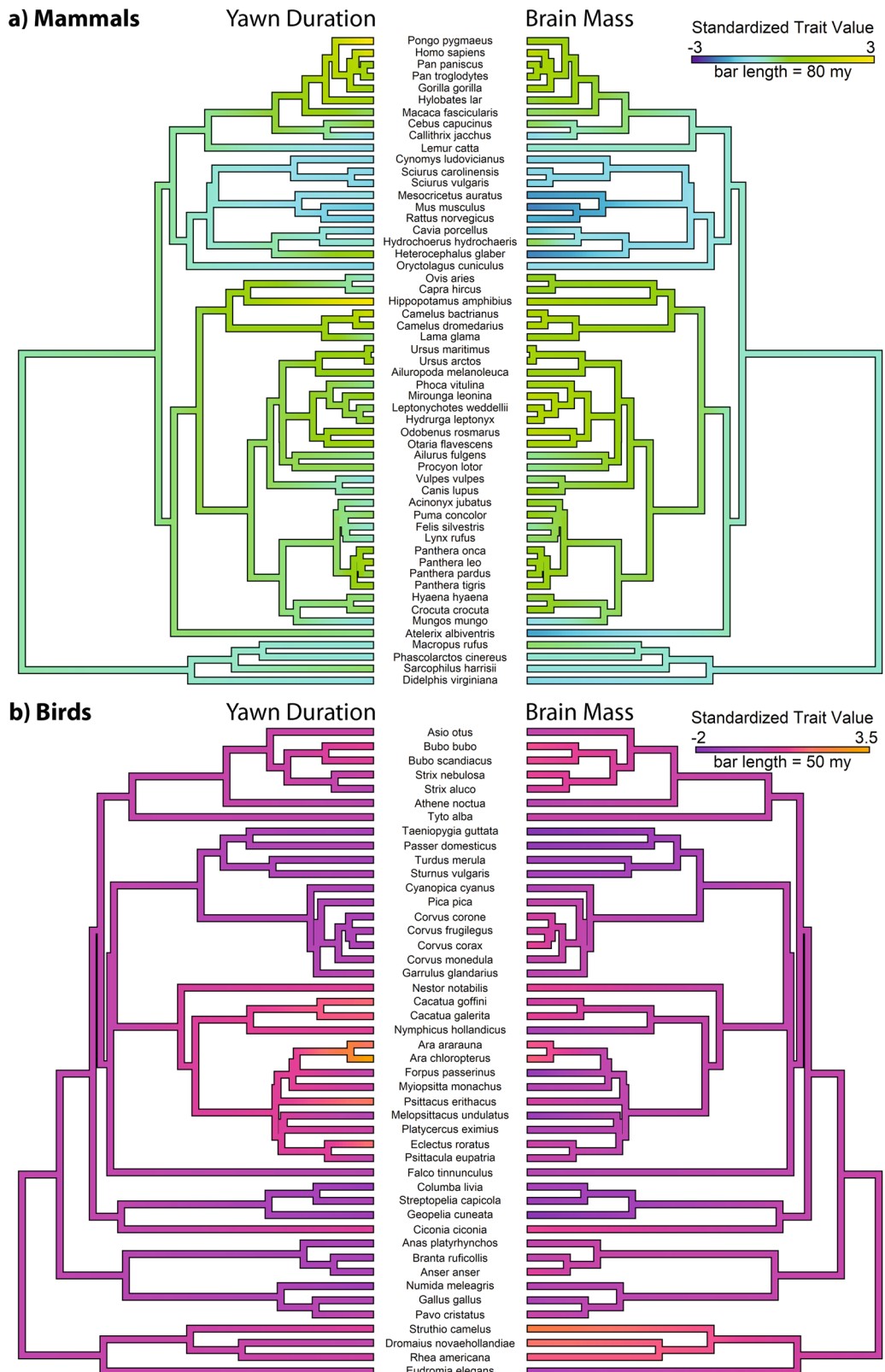

**Fig. 2 Yawn duration and brain weight across mammals and birds.** Standardized measures of species-typical yawn duration (left) and log-scale absolute brain weight (right) are painted across mammalian (**a**) and avian (**b**) phylogenetic lineages. Z-score standardized values are presented to facilitate comparison between traits.

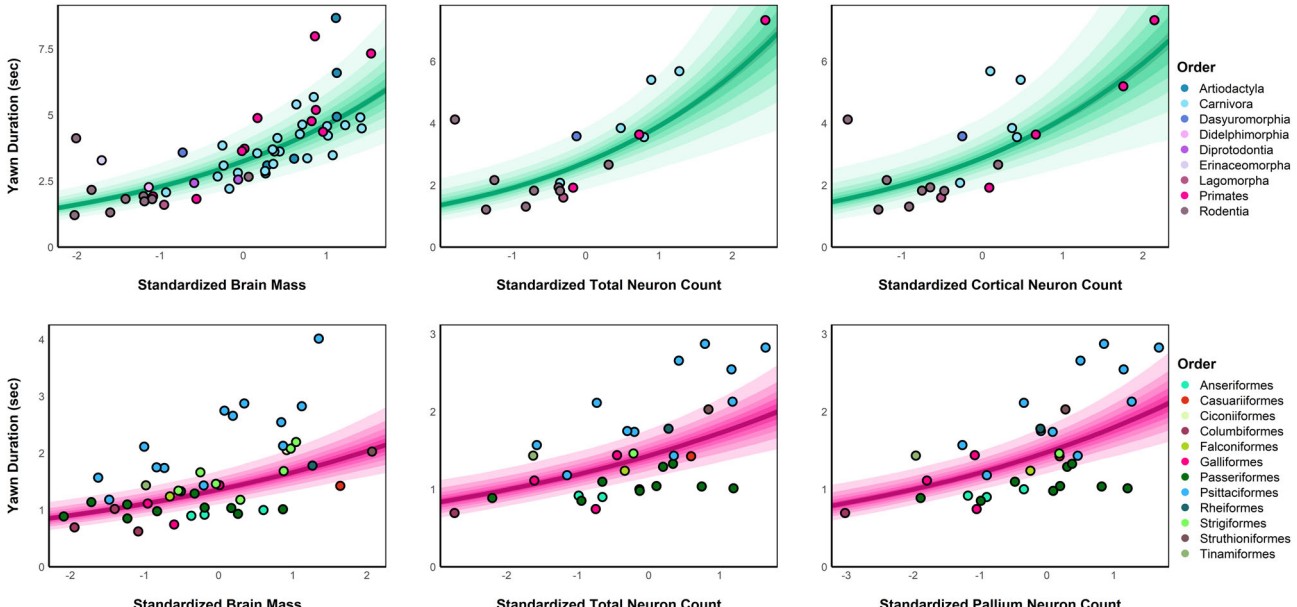

**Fig. 3 Predicted associations between brain measures and yawn duration across taxa.** Model predictions are shown for the relationship between each neurological measure and the expected yawn duration across mammals (green slopes) and birds (pink slopes), marginalizing over species-level random effects. Y-axes are adjusted to the range of observed values across measures due to smaller sample sizes and a subsequently reduced range for the neuron count data. Posterior median values are indicated by the dark line, with posterior uncertainty represented by shaded bands of 10–90% credible intervals from darkest to lightest, respectively. Observed species-level values are shown as circles colored by taxonomic order.

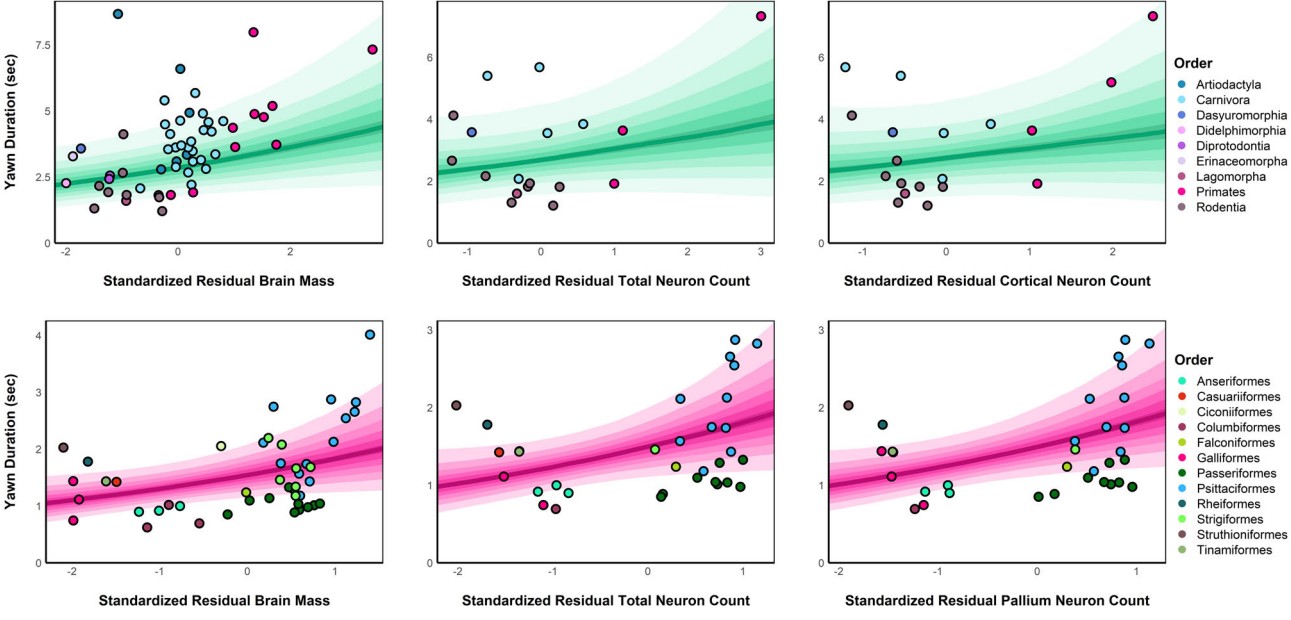

**Fig. 4 Predicted associations between residual brain measures (controlling for body size) and yawn duration across taxa.** Model predictions are shown for the relationship between the residuals of each neurological measure and the expected yawn duration across mammals (green slopes) and birds (pink slopes), marginalizing over species-level random effects. Y-axes are adjusted to the range of observed values across measures due to smaller sample sizes and a subsequently reduced range for the neuron count data. Posterior median values are indicated by the dark line, with posterior uncertainty represented by shaded bands of 10–90% credible intervals from darkest to lightest, respectively. Observed species-level values are shown as circles colored by taxonomic order.

sizes for neuron counts and high correlations among brain measures (see Supplementary Tables 1 and 2), which indicate that the size and count measures are providing largely redundant information across taxa.

Finally, we also ran models on neuronal density, as a potential metric of heat generation per brain volume unit, yet did not find any clear patterns after controlling for body weight,

i.e., when taking residual brain neuronal density measures (see Supplementary Note 4).

**Mammal and bird yawns compared**. After correcting for phylogenetic signal, we found little evidence for differences between mammals and birds in the effect size of absolute brain mass

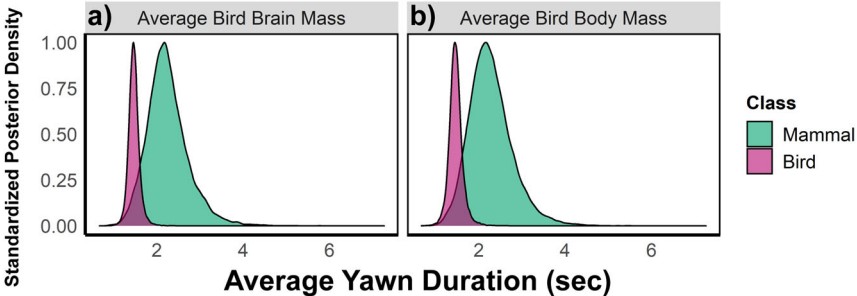

**Fig. 5 Comparing the yawn durations of mammals and birds.** Posterior predictions for the average mammal's (green) and bird's (pink) yawn duration at the average brain size (**a**) and body size (**b**) of birds in our sample, controlling for phylogeny.

($\Delta\beta = -0.06$ [0.06], 90% CI [−0.18, 0.04], $p_+ = 0.15$), total neuron count ($\Delta\beta = -0.04$ [0.08], 90% CI [−0.17, 0.09], $p_+ = 0.31$), or cortical and pallial neuron count ($\Delta\beta = -0.02$ [0.08], 90% CI [−0.15, 0.10], $p_+ = 0.40$). Our data therefore do not provide support for a difference in the slopes of yawn duration on either brain size or complexity between mammals and birds (see also Supplementary Fig. S1). However, mammals in our sample exhibited a considerably longer average yawn duration than birds ($\Delta\beta_0 = 0.85$ [0.21], 90% CI [0.49, 1.22], $p_+ = 0.99$), with an expected yawn duration of 3.40 and 1.46 s for mammals and birds of average brain size within their respective clades. Furthermore, our models predicted a longer yawn duration for mammals even at the same brain ($\Delta\beta_0 = 0.41$ [0.21], 90% CI [0.04, 0.78], $p_+ = 0.97$; Fig. 5a) and body size ($\Delta\beta_0 = 0.43$ [0.23], 90% CI [0.04, 0.83], $p_+ = 0.96$; Fig. 5b) of the average bird. Thus, despite common scaling patterns, our data suggest that the difference in average yawn duration between mammals and birds cannot be solely attributed to allometric scaling with size.

## Discussion

Our data show a clear relationship between brain mass and neuron numbers and the duration of yawning in both mammals and birds that cannot be explained by allometry alone. Consistent with previous work on mammals only and on a much smaller sample[15–17], our data imply that yawning is an adaptation with an important neurophysiological function. Moreover, this function seems to be conserved across a diverse range of animals, such that its evolutionary origin may be traced back to at least the common ancestor of birds and mammals and potentially even further. Whereas our data do not speak to the precise neurophysiological function of yawning, in combination with previous studies related to thermoregulation[4–14,18–20], a brain cooling function seems most probable.

Based on the brain cooling hypothesis, the extended gaping of the jaw combined with the deep inhalation of air that characterizes yawning functions to cool the brain by altering both the rate and temperature of the arterial blood traveling to the brain[4–6]. Accordingly, the thermal changes from yawning should be tied to the duration or magnitude of this response. Since increases in brain mass produce greater thermolytic needs[27], and brain temperature is determined in part by metabolic heat production from neurons[28], animals with larger and more complex brains would require longer and more powerful yawns to achieve the same functional outcomes. Consistent with this prediction, phylogenetically controlled analyses from over 100 species of mammals and birds revealed robust positive correlations between yawn duration and absolute brain mass, and with total and cortical/pallial neuron numbers. Moreover, brain mass remained a significant predictor of interspecies yawn duration across both

mammals and birds even after correcting for body size. While deviations in total brain ($N = 18$) and cortical ($N = 19$) neuron numbers from expected body mass did not provide a clear relationship with yawn duration in mammals, significant positive associations of moderate size were still observed for both total and pallium neuron numbers ($N = 33$) in birds.

However, it should be noted that our primary hypotheses concern the evolution of yawn duration in response to brain size and neuron numbers, irrespective of whether these neurological measures evolve through direct selection or indirectly through selection on body size. Given these considerations, we therefore conducted our primary statistical analyses without including body weight as an additional covariate. While controlling for body size is crucial to ensure that the observed effects of brain size on yawn duration are not solely attributable to allometric scaling, this procedure also significantly reduces the variation in absolute brain measures which we hypothesized to predict yawn duration. In other words, by investigating brain size variation independent of body size, we are in essence testing a distinct biological hypothesis—i.e., whether *deviations* in brain size from phylogenetically expected body size predict yawn duration—rather than our primary hypothesis with regards to the physiological consequences of *absolute* brain size (irrespective of whether this brain size is expected given body size). Nonetheless, we still find clear effects of body size adjusted brain size on yawn duration in both mammals and birds, as well as for body size adjusted neural counts (both total and pallial) in birds, demonstrating the robustness of our findings. The increased statistical uncertainty observed for the nearly equivalent effects of body size adjusted mammalian neuron counts (both total and cortical) can also be clearly attributed to the loss of variability and statistical power caused by partialling out body size in the relatively small samples available for those measures ($n = 18$ and 19 respectively). Indeed, phylogenetic correlations were very high between all brain measures (see Tables S1 and S2), suggesting that the total brain size and neuronal count measures are providing nearly equivalent information about the evolution of yawn duration in our analyses. Future studies would benefit from investigating these associations within lineages where neuron counts and total brain size exhibit more independent variation than observed in the present study, which would aid in parceling out distinct causal effects of brain size and neuron counts on yawn duration.

The overall pattern between yawn duration and the neurological measures was robust across both vertebrate classes, as there was no clear difference in the slopes between mammals and birds (see Fig. 3). We did, however, find that the average bird yawn was significantly shorter than that of mammals, and that, even at a comparable brain and body mass, mammals are expected to yawn longer on average. This finding is consistent with the purported brain cooling function of yawning since body temperature, and

consequently the temperature of blood, is approximately 2 °C higher in birds than in mammals[38]. As a result, heat exchange between blood and atmosphere is faster in birds than mammals, and thus birds would not be required to yawn as long to achieve the same cooling effect. As mentioned above, birds also have a morphological adaptation, the rete mirabile ophthalmicum, that allows for selective brain cooling[29,30]. While this system would still benefit from behaviors like yawning for the supply of cooler arterial blood, its increased efficiency may not require the same magnitude of response compared to mammals. Nevertheless, a similar adaptation, the carotid rete, has been observed in several mammal orders; i.e., artiodactyls and felids and rudiments of this structure also in canids[39,40], and these orders do not seem to show shorter yawns than other mammalian orders (see Fig. 3).

Beaks are another site of heat loss specifically in birds that has a direct connection with nasal and oral cavities[30], and in toucans, for example, rates of heat loss reach 400% of resting heat production[41]. As a consequence, birds would not require long yawns due to more effective forms of heat dissipation. Variation in beak size may then be a powerful tool to investigate the residual variation in our data. We do find longer yawns in species with relatively shorter beaks such as in owls and parrots. Although, with regard to the latter it is difficult to disentangle this potential effect from the parrots' large relative brain size and their high neuronal density in the brain[42,43], which are the main effects demonstrated in this study. Moreover, the curved beaks of both owls and parrots may require them to open further and consequently longer while yawning for the same heat exchange. In sum, future studies on the thermoregulatory function of yawning should take into account specific morphological differences within and between clades.

Finally, following from the high neuronal densities, it is likely that avian neurons are on average much smaller, with less extensive dendritic arbors than mammalian neurons[42]. This is an important factor that reduces the energy consumption of an avian neuron compared to an average mammalian neuron. Indeed, it has been recently shown that an average neuron from the pigeon is 3-fold energetically cheaper than an average mammalian neuron[44,45]. These results strongly suggest that despite high neuronal densities, avian brains may actually produce less heat than mammalian brains of equal size, which would explain why they display shorter yawns even at the same brain and body size.

When examining the data, a distinct outlier was present within the mammalian sample: the naked mole-rat (Heterocephalus glaber) (see Fig. 3). This species displayed very long yawns (4.12 s) given their relatively small brain mass and neuron numbers[46]. By comparison, the yawn duration of naked mole-rats was equivalent to those of jaguars (4.13 s), which have a brain that is approximately 300 times as large. Naked mole-rats are unique among mammals in that they are poikilothermic, relying on the environment and behavioral methods to control heat loss[47]. Yet, naked mole-rats lack sweat glands[48] and do not pant or increase salivation during heat stress[49]. As a result, further studies could examine whether yawning serves more of a primary mechanism for regulating brain temperature in this species, which could explain their disproportionately long yawns. Given the eusociality of naked mole-rats, further research could also examine the potential social function(s) of yawning in this species. Similarly, extending the current line of research to ectothermic clades, which are known to yawn[2] and potentially benefit even more from behavioral thermoregulatory adaptations, would be an important future direction.

In conclusion, phylogenetically controlled analyses across a large and representative sample of mammals and birds show that although the pattern of yawning is fixed, its duration has co-evolved with brain size and neuron numbers. Moreover, these

findings provide further support for distinct predictions derived from the brain cooling hypothesis, and show that this complex reflex is highly conserved across taxa and can likely be traced back to at least the common ancestor of mammals and birds.

## Methods

**Yawn measures.** Data collection took place between March 2017 and December 2019. Videos of yawning mammals and birds were collected from online sources (YouTube, shutterstock, gettyimages, footage framepool, vine, 123rf, istockphoto), from videos provided by colleagues or zoos, or were collected by the authors using handheld cameras filming the different animals in zoos and research institutes (see "Acknowledgement" section). Data on individual yawns, including the source and the URL's are available in an online depository (see "Data availability" section). Videos shot by the research team are available in full in an online depository (see "Data availability" section). Video's of colleagues and zoos can be made available upon request. In total we collected 1557 yawns (831 mammal yawns and 726 bird yawns), from 810 individuals (523 mammals, 287 birds) across 110 species (60 mammal species and 50 bird species). MH compiled all videos. All videos were then checked for validity (is it a yawn yes/no; and, are there any clear (social) triggers that may have caused the yawn, thereby making it a non-spontaneous yawn), and species identity by JV, and all doubtful cases were subsequently checked by ACG. Furthermore, we excluded all species of which we did not have at least two different yawns; i.e., also from two different individuals. This reduced our sample to 1504 yawns (811 mammal yawns and 693 bird yawns), from 787 individuals (507 mammals, 280 birds) across 108 species (60 mammal species and 48 bird species).

MH coded all videos using the trim function of QuickTimePlayer (Apple) for yawn duration following the definition of a yawn as provided by Barbizet[1]: "Yawning is a wide gaping of the mouth with a deep involuntarily breath with a peak muscle contraction, followed by exhaling and the closing of the mouth." In this study, we operationalized this as follows: the start-moment of the yawn was the last moment of the closed mouth (mammals: regarding their lips) or bill (birds: regarding the tip of their bill) and the end-moment was the moment of the again closed mouth or bill. If the bill or mouth was not closed before the yawning event, the last moment before the movement to open it further was used, and similarly if the mouth or bill was not completely closed in the end, the moment of the stop of the closing movement was used. A random selection of 16.3% of all videos were recoded by EL and inter-rater reliability was excellent (Spearman's rho = 0.9718691, $p < 2.2e-16$), also when looking at the two vertebrate classes separately (Mammals: 16.9% of all videos: Spearman's rho = 0.9110349, $p < 2.2e-16$; Birds: 15.5% of all videos; Spearman's rho = 0.971419, $p < 2.2e-16$). Note that some of the available videos had been analysed for their duration in previous studies[26–28]. Therefore we also calculated inter-rater reliability between MH on all videos that were used in those previous samples and the current one (18.8%), and again inter-rater reliability was excellent (Spearman's rho = 0.914544; $p < 2.2e-16$).

**Brain measures.** Body mass and brain mass/endocranial volume (ECV) of all species were extracted from the literature. ECV was converted to brain mass using the equation brain mass = $1.036 \times ECV$[50]. We calculated the average brain mass and body mass for each species, and the inter-sex mean when separate values for males and females were available. Data of unstated sample size were treated as single individuals. Similarly, total number of brain neurons and number of cortical/pallial neurons for 18/19 (total and cortical respectively) mammals and 23 species of birds were extracted from the literature[42,46,51–58]. Neuron counts for additional ten avian species were newly estimated using the isotropic fractionator[36]. Briefly, two to four individuals per species were collected, with exception of the Red-breasted goose (Branta ruficollis), in which only one individual was examined. Animals were killed by an overdose of halothane. They were weighed and immediately perfused transcardially with warmed phosphate-buffered saline containing 0.1% heparin followed by cold phosphate-buffered 4% paraformaldehyde solution. The brains were immediately removed, weighted, postfixed for an additional 7–21 days and then dissected into five brain divisions, namely the telencephalon (cerebral hemispheres, including the olfactory bulbs), diencephalon, optic tectum, cerebellum, and brainstem. In one individual per species, one hemisphere was dissected into the pallium and the subpallium. Dissection procedures were described earlier[42]. The dissected structures were dried with paper towel, weighed to the nearest milligram, incubated in 30% sucrose solution until they sank, then transferred into antifreeze (30% glycerol, 30% etylene glycol, and 40% phosphate buffer) and frozen for further processing. All procedures were approved by Institutional Animal Care and Use Committee at Charles University in Prague, Ministry of Culture (Permission No. 47987/2013) and Ministry of the Environment of the Czech Republic (Permission No. 53404/ENV/13-2299/630/13).

The dissected brain parts were homogenized in 40 mM sodium citrate with 1% Triton X-100 using Tenbroeck tissue grinders (Wheaton, Millville, NY, USA) to obtain a suspension of free cell nuclei. The fluorescent DNA marker 4′,6-diamidino-2-phenylindole (DAPI) was added (0.5 mg/l) to stain the nuclei, homogenate was adjusted to defined volume and the mixture was kept homogenous by agitation. The total number of cells was estimated by counting at least five aliquots of 10 μl using a Neubauer improved counting chamber (BDH,

Dagenham, Essex, UK) with an Olympus BX51 microscope equipped with epifluorescence and appropriate filter settings; additional aliquots were counted when needed to reach the coefficient of variation among counts ≤0.10. The proportion of neurons was determined by immunocytochemical detection of the neuronal nuclear marker NeuN[59]. This neuron-specific protein was detected by an anti-NeuN mouse monoclonal antibody (clone A60, Sigma-Aldrich; dilution 1:800), which was characterized by Western blotting with chick brain samples and shown to react with a protein of the same molecular weight as in mammals[60], indicating that it does not cross-react with other proteins in birds. The binding sites of the primary antibody were revealed by a secondary anti-mouse IgG antibody conjugated with Alexa Fluor 594 (Life Technologies, Carlsbad, CA, USA; dilution 1:400). An electronic hematologic counter (Alchem Grupa, Torun, Poland) was used to count the proportion of double-labeled nuclei in the Neubauer chamber. At least 500 nuclei were examined for each sample.

All data on brain measures per species, as well as the sources of those data are available in an online depository (see "Data availability" section).

As brain measures for dogs and chickens tend to vary substantially per breed[61,62] we decided to exclude these two species from further analyses, reducing our sample with another 40 yawns and 22 individuals. Furthermore, note that total neuron number and neuron numbers of the cortex/pallium were not yet known for all species in our sample and analyses on these measures were thus on a restricted sample (total neuron number, $n = 18$ and 33 species for mammals and birds, respectively; neuron numbers of the cortex/pallium, $n = 19$ and 33 species for mammals and birds, respectively).

## Analyses

*Brain measures.* Our species-level measures of average brain mass, body mass, total neuron count, and cortical and pallium neuron counts were highly skewed. We therefore log-transformed these measures prior to statistical analysis. On the log scale, our brain and body mass measures were highly correlated ($r = 0.95$ for mammals, $r = 0.87$ for birds), as were body mass and the mammalian total neuron count ($r = 0.88$) and cortical neuron counts ($r = 0.82$). In contrast to mammals, avian body mass had smaller associations with total neuron count ($r = 0.51$) and pallium neuron count ($r = 0.47$). Please see Supplementary Tables S1 and S2 for similar estimates of the phylogenetic correlations between all measures.

Our primary hypotheses concern the evolution of yawn duration in response to brain size, irrespective of whether brain size evolves through direct selection or indirectly through selection on body size. We thus first investigated the overall association between yawn duration and our brain measures without adjusting for body size. However, because of the strong associations between brain and body size (see Table S1–2) in our sample, spurious associations may arise due the evolution of both brain size and yawn duration in response to changing body size, rather than any direct causal relationship between these traits. We therefore conducted further analyses adjusting for body size to examine whether these associations also reflected unique relationships between brain size and yawn duration, rather than allometric scaling with body size alone.

It is common to adjust for body size in comparative studies of brain evolution by including body weight as an additional predictor in a multivariate analysis. However, recent work strongly suggests against this practice due to the high risk of inferential bias from multicollinearity[63]. When the risk of multicollinearity bias is high, it is suggested that phylogenetically controlled residuals should instead be utilized for identifying unique evolutionary relationships with relative brain size[64]. To address this concern, we first used variance inflation factors (VIFs), which provide a standardized metric of multicollinearity, to quantify the risk of bias in multivariate models including brain and body size measures as predictors of yawn duration. VIFs of 1 indicate no multicollinearity, while VIF > 3 have been suggested to indicate undesirably high risk of multicollinearity bias in ecological datasets[65].

As expected given the raw and phylogenetic correlations (see Tables S1–S2), VIFs were undesirably high for the mammalian total brain size (VIF = 13.05) and total neuron count measures (VIF = 5.10), as well as for the avian total brain size measure (VIF = 3.59). In contrast, VIFs were sufficiently low for the mammalian total cortical neuron count (VIF = 2.91), as well as for the avian total neuron count (VIF = 1.29) and total pallial neuron count (VIF = 1.06). To ensure reliable inference for measures with VIF > 3 (i.e., mammalian total brain size and total neuron count, avian total brain size), we therefore calculated residual brain measures from a phylogenetically controlled linear regression of the respective brain measure on body mass[37], and used these residuals as predictors in subsequent analyses. For measures with VIF < 3 (i.e., mammalian total neuron count, avian total neuron count and pallium neuron count), we instead included body size as an additional predictor in a multivariate regression analysis.

*Phylogeny.* In order to account for autocorrelation due to phylogenetic history, we utilized mammalian and avian phylogenetic trees available for the species in our sample[66,67] (see also Fig. 1). Phylogenetic trees specific for this sample are available in an online depository (see "Data availability" section). As domesticated species have experienced different evolutionary constraints, the length of their phylogenetic distances is unclear and it is therefore difficult to correctly

place them in a phylogenetic tree. As a consequence, we had to exclude an additional five species (four mammals, one bird), leading to the final sample as described in the main text. Within this final sample, moderate to large phylogenetic signals were estimated for yawn duration in mammals ($\lambda = 0.80$) and birds ($\lambda = 0.64$).

*Statistics and reproducibility.* We used Bayesian multilevel phylogenetic models to investigate the associations between yawn duration and the brain measures while accounting for species' phylogenetic history. These models extend and provide a number of additional benefits in comparison to classical phylogenetic least squares methods[68,69]. Among others, these include (i) allowing for phylogenetic analysis of non-Gaussian responses, such as our yawn duration measure; (ii) estimation of random effects accounting for hierarchically structured data, as is necessary for our repeated measures across species; and (iii) the use of regularizing priors that produce more conservative predictions and reduce the risk of inferential error[70,71]. Due to multicollinearity and the absence of neuron count measures for some taxa, we estimated separate models for the association between each brain measure and the average yawn duration sampled from multiple individuals of each species (mean = 7.75 [SD = 5.64] individuals per mammalian species, and mean = 5.89 [SD = 4.90] individuals per avian species). Mammalian and avian data were analyzed separately to enhance the accuracy of phylogenetic inference.

For each model, we treated yawn duration as a Gamma distributed response variable with a log link function, which appropriately accounted for the non-negative, continuous nature of this duration measure[72]. Each species-level brain measure was included as a fixed effect to address our primary hypotheses. Values were standardized to $z$-scores (i.e., [value − mean]/SD) to facilitate effect size comparison across brain measures. In addition to phylogenetically structured random effects, we further included independent species-level random intercepts to account for any unobserved heterogeneity among species not explained by phylogenetic history. In addition, we estimated random slopes models to investigate whether associations between yawn duration and the brain measures differed across mammalian and avian orders. We used the Watanabe–Akaike information criterion (WAIC), a fully Bayesian extension of classical AIC accounting for model uncertainty[73], to compare models with and without these order-level effects. As with standard AIC, $\triangle$WAIC = 2 provides minimal evidence for selecting the more complex random slopes model over a model with a fixed slope across orders.

In addition to estimating associations between yawn duration and neurological measures within our mammal and bird samples, we also formally compared the intercept of yawn duration and slopes of the brain size effects between these samples. This allowed us to assess whether the average yawn duration and the effect of brain mass/neuron numbers on this behavior differed between mammals and birds, while still appropriately correcting each estimate for phylogenetical autocorrelation within the respective samples. In particular, we compared intercepts and slopes by quantifying the difference between the posterior distributions of each parameter for mammals and birds (i.e., $\triangle\beta_0$ for intercepts and $\triangle\beta$ for slopes). As reported in the results section, we found clear differences in the intercepts of the average mammal and bird yawn duration, but little evidence for differences in the slopes of brain size effects. Given that the mammal and bird samples differed in their mean absolute brain and body mass, the difference observed in their mean-centered intercepts could merely reflect allometric scaling of yawn duration with size. We therefore fit additional models in which the brain and body size slopes of mammals and birds were constrained to equivalence during estimation, which we used to predict whether the average yawn duration would be expected to differ between mammals and birds of the same brain and body size.

Regularizing $\beta \sim \text{Normal}(0, 1)$ priors and $\sigma \sim \text{Exponential}(3)$ priors were placed on fixed and random effects respectively[74] in all analyses. The "brms" package for R statistical environment[75] was employed for all analyses, which interfaces with the Stan statistical programming language[76]. In contrast to classical methods, which rely on point estimates and the well-known pitfalls of null hypothesis testing[77], Bayesian statistics produce a probability distribution capturing uncertainty in each estimated parameter, which are known as posterior distributions. Following state-of-the-art statistical practice[70,71,77], we used multiple quantitative measures to summarize these posterior distributions and draw scientific inferences from our models. In particular, to interpret the strength and uncertainty of estimated regression effects, we present the posterior median regression coefficient ($\beta$), the median absolute deviation as a robust measure of posterior dispersion (presented in brackets), 90% Bayesian credible intervals (CI), and the proportion of the posterior probability greater than 0 ($p_+$). Note that in contrast to classical $p$-values, which indicate the probability of observing the data under a null hypothesis $p(\text{data}|\text{H}_0)$, the reported $p_+$ directly estimate the probability of positive associations between yawn duration and each brain measure after observing our data $p(\text{H}_1|\text{data})$. Values of $p_+$ closer to 1 therefore indicate stronger support for a positive effect. In addition, we calculated Cohen's $d$ standardized effect sizes to facilitate comparison within and across studies, with values of $d \geq 0.2$, 0.5, and 0.8 interpreted as small, medium, and large effects respectively[78].

Code as well as csv data-files are available in an online depository (see "Data availability" section).

**Reporting summary**. Further information on research design is available in the Nature Research Reporting Summary linked to this article.

## Data availability

All data are made available on an online depository; i.e., DataverseNL. This "Dataverse" contains: 1. Source data for brain measures with additional references, as well as source data for each yawn, separately for mammals and birds. (https://doi.org/10.34894/ROFNL1). 2. Database of all yawn videos collected by the authors, or with permission from original recorder(s) (birds; https://doi.org/10.34894/GYTSEK; mammals; https://doi.org/10.34894/XJQBLB). 3. Phylogenetic trees for both mammals and birds, specific to the samples of our analyses. (https://doi.org/10.34894/K9SIZF). 4. Final data files (csv). (https://doi.org/10.34894/9HENTF)

## Code availability

R scripts. (https://doi.org/10.34894/KT5DTY).

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

## Acknowledgements

We are very grateful to Christian Blum, Christiane Rössler, Katie Slocombe, Sita ter Haar, Raoul Schwing, Auguste von Bayern, Jingzhi Tan, Evy van Berlo, Alejandra Díaz-Loyo, and Oscar Juárez-Mora for providing us with videos of their yawning animals. Similarly, we thank Avifauna, Tierpark Stadt Haag, Zoo Linz, Haidlhof Research Station, Konrad Lorenz Forschungsstelle Grünau and Wildtierpark Cumberland, Nationalpark Thayatal, Zoo Bratislava, Eulen- und Greifvogelstation Haringsee, Papageienschutzhaus and Tierschutzheim Vösendorf, and Tiergarten Schönbrunn and in particular Jonas Küh-napfel, for allowing us to collect videos of yawns of the animals in their collection. Part of this research was funded by the Austrian Science Fund (FWF, P 26806 to J.J.M.M.), the Czech Science Foundation (18-15020S, to P.N.), and the Grant Agency of Charles University (1438217 to M.K.).

## Author contributions

J.J.M.M. and A.C.G. conceived the project. J.J.M.M., M.H., E.L., J.H., and A.C.G. collected yawn data. M.K., S.O., C.O., Y.Z., and P.N. collected and compiled brain data. M.H. coded all yawn data, with oversight from J.J.M.M., E.L., J.V., and A.C.G. J.S.M. conducted the statistical analyses with input from J.J.M.M., P.N., and A.C.G. J.J.M.M., M.H., J.S.M., P.N., and A.C.G. wrote the manuscript and supplemental materials, with oversight from all other authors.

## Competing interests

The authors declare no competing interests.
