## [Peer Review File · Communications Biology]

Reviewers' comments:

Reviewer #1 (Remarks to the Author):

The manuscript by Massen et al. tests an interesting hypothesis that longer yawns occur in larger brained mammals and birds due the importance of yawning in regulating brain temperature. The data does indeed support an evolutionary correlation between brain size and yawn length. I do have a few questions about the data and the analysis, although well executed, needs to also consider allometric effects (irrespective of collinearity concerns as I discuss below).

1. One aspect of yawning that I felt was overlooked was its potential use as a social signal. Are there any quantitative differences in socially directed yawns as opposed to 'normal' or 'typical' yawns? This information would be helpful in assessing the behavioural data that was acquired as there might be circumstances where exaggerated yawns occur that do not serve brain cooling. Similarly, is there any data on yawning frequency or duration in relation to stress/anxiety? What I am trying to get at here is that the authors explicitly set this up with respect to the brain cooling hypothesis and while I am not denying the importance of that hypothesis, the other functions of yawns might also be important when considering the evolution of yawn length in a comparative context.

2. The authors raise the issue of multicollinearity and suggest that this is a strong rationale for not examining relative brain size in their study. Multicollinearity can be an issue in studies of relative brain size and this is true of any variable that exhibits allometry. As far as I am aware, there is still no effective means of addressing this in a comparative context, but I do not agree that this means that it should not be tested at all. Further, by not including body size in all of their analyses, the authors are potentially ignoring an allometric effect on yawn duration. For example, all of the graphs in Figure 3 look like exponential (i.e., allometric) relationships that could be driven by body size and not brain size or neuron numbers. The brain:body ratio as a means of controlling body size is also not an effective means of incorporating overall size; ratios are often correlated with their denominators in evolutionary allometry. Thus, the brain:body ratio is probably correlated with body size, potentially indicating that body size is indeed exerting an effect on yawn duration. I would therefore suggest the following analyses: 1) examine body size explicitly in all comparisons to determine if similar relationships occur across all variables; and 2) include body mass in models of brain size, neuron number, etc. and acknowledge that collinearity can affect the statistical modeling.

3. In addition to the analyses I suggest above, the authors should examine neuronal density. Neuronal density is inversely correlated with brain size (and brain region size) and could provide further insights into the relationship between the brain and yawn duration.

4. The Discussion places a lot of emphasis on the brain cooling hypothesis, but it remained a bit unclear to me why a correlation between brain size and yawn duration would be necessary. If the primary function of yawning is to cool the brain, does this imply that larger brains have greater heat dissipation needs? Or is it that larger bodies generate and retain more heat and therefore might require longer yawns in order to cool the larger brains present in larger animals? Including the analyses I suggested would address these questions.

5. Is it possible that the extended yawns in the naked mole-rat are unrelated to thermoregulation of the brain and reflect some other function of yawning? It is a highly social species, so I was thinking about the possible use of yawns as a social signal or to reflect stress/anxiety.

6. I am not sure that I agree with the authors that the data supports a brain cooling function based on what was presented. There is little doubt that brain size and neuron numbers have evolved in concert with longer yawns, but without controlling for body size, this could reflect allometry or something else. I think it certainly points to an interesting example of correlated evolution, but the link to the brain cooling theory needs to be explained in a more explicit fashion.

Reviewer #2 (Remarks to the Author):

In this paper the authors perform a comparative study aimed at testing the 'brain cooling hypothesis' for yawning, by examining correlations between brain size, other neural traits and yawn duration. I was interested to read the paper having read previous comparative analyses from some of the same authors. I didn't find these previous analyses convincing due to the modest sample sizes (in most cases) and some statistical issues, in particular the lack of phylogenetic correction (ref 15), use of multiple correlated traits and ratios (refs 15, 16), and the use of brain size estimates derived from transformations of body size (ref 17). Indeed, the authors acknowledge some of these issues on lines 75-77, which is good to see. In any case, it was interesting to see the topic revisited with larger samples and potentially more robust methods.

Unfortunately, although the paper is nicely written and the Bayesian analyses seem to be well implemented, I still think some of these basic concerns remain. In particular:

1. No evidence is presented that the associations are specific to brain mass, or any other neural trait – I'd repeat the analyses with body mass to compare fit. If you can't find evidence brain mass predicts the yawn data better than body mass I don't think your conclusions are justified.
2. Ratios are not a valid measure of relative brain size as brain and body mass does not scale isometrically.
3. The results for brain mass, neuron number, cortical neuron number are treated as if they are independent of each other, but they are not. As such, they don't provide independent support for the hypothesis.

As an aside, none of the dataverse links worked for me (I get a 'page not found' error) so I have been unable to look at these in detail.

Specific points in more detail:

- The collection of the yawn data is well described and seems as robust as it can be. I'd like to see more detail on how species identity was made for YouTube (and other public) videos. I'd also point out that many YouTube links are not stable, for example in some of the previous papers the links no longer work. But I'm not sure how this can be accounted for, unless the authors download all the videos, but that seems a bit arduous...
- I think it is justified to analyse brain mass without including body weight in the analysis if your hypothesis concerns absolute size, provided the same analyses are also performed with body mass allowing direct comparisons of model fit. If the brain models fit better than the body models you can claim a main effect of brain mass, if not, you can't rule out general associations with body mass and I don't think your inferences would be supported in that case. The neuron number data may be your best bet here, since neuron number varies with body mass less consistently than brain mass. If you can't test for independent effects of neuron number and body mass then I would be pretty unconvinced.
- In addition, if you're justifying not including body mass in the analyses due to the effects of collinearity you should probably test that this is actually happening in your data. Alternatively, you can potentially try to side step the problem to some extent by regressing both yawn duration and brain mass against body mass and then calculate and compare the residual values to check that variation in these traits that is independent of body size are correlated, as your argument would predict. This is not ideal as residuals are not data, but it would provide another way to bolster your conclusion and might be a useful supplementary analysis if collinearity is a problem.

- I don't agree brain/body ratios are statistically valid measures of relative brain size as they do not appropriately correct for brain~body allometry. These analyses should be replaced with multiple regressions (e.g. a basic model of yawn ~body mass + brain mass). I don't believe any analysis based on ratios is robust when the traits involved do not scale isometrically. The ratios will probably still correlate with body size.

- Including multiple neural traits in the analyses is complicated by the fact that they are all correlated with each other, and therefore not statistically independent. In this context it's interesting the beta estimates tend to be very similar across traits within a dataset. You don't really compare the results with brain mass or neuron number, but they are presented as if they provide independent lines of support for your hypothesis. Instead, you could use these data to confirm the most meaningful trait, either by comparing fit of models with different traits, for example. When comparing models obtained with different traits they should be based on the same sampling of species, so you would need to subsample the brain dataset and repeat the analysis to compare it to the neuron number dataset, for example. Regression models can also be built that narrow down the effect, e.g. is the effect due to neuron number, independent of the effects of brain size, or vice versa which might suggest non-neuronal cells are important.

- Another trait I would like to see included in the analyses is lung mass or volume, as there (to me) it seems reasonable to expect that yawn duration is constrained (or correlated) with lung capacity. There is data available for mammals at least, in Navarrete, A., van Schaik, C. P., & Isler, K. (2011). Energetics and the evolution of human brain size. *Nature*, 480(7375), 91-93.

- Can the authors report the strength of phylogenetic signal, and how this was incorporated in the analyses? Do the models assume high/complete phylogenetic signal? Or is it estimated at the same time as the other parameters.

- Line 529-530: can you show that it does so, rather than just saying it does? Similarly, how was the gamma distribution detected?

- Figure 2: I find these figures a little uninformative, if you replace the plots of brain mass with plots of body mass they would look almost identical.

In sum, I think this is an interesting topic, and the authors explain the logic behind the analyses very well. I just don't think they explore the data sufficiently to make the argument that these associations have anything to do with brain size specifically. As it currently stands my interpretation of the results would simply be that bigger birds/mammals have longer yawns. However, I do think there are ways of analysing and supplementing these data to make the authors' argument more robust and I hope the suggestions above help.

Reviewer #1 (Remarks to the Author):

The manuscript by Massen et al. tests an interesting hypothesis that longer yawns occur in larger brained mammals and birds due the importance of yawning in regulating brain temperature. The data does indeed support an evolutionary correlation between brain size and yawn length. I do have a few questions about the data and the analysis, although well executed, needs to also consider allometric effects (irrespective of collinearity concerns as I discuss below).

1. One aspect of yawning that I felt was overlooked was its potential use as a social signal. Are there any quantitative differences in socially directed yawns as opposed to 'normal' or 'typical' yawns? This information would be helpful in assessing the behavioural data that was acquired as there might be circumstances where exaggerated yawns occur that do not serve brain cooling. Similarly, is there any data on yawning frequency or duration in relation to stress/anxiety? What I am trying to get at here is that the authors explicitly set this up with respect to the brain cooling hypothesis and while I am not denying the importance of that hypothesis, the other functions of yawns might also be important when considering the evolution of yawn length in a comparative context.

While yawns may serve a derived social function in some social species (and we now discuss this in relation to the naked mole-rats below), the primitive form of yawning is clearly physiologic and we are not aware of any socially directed yawns that occur across birds and mammals. Although "threat yawns" have been described as a signal among some non-human primates, these represent canine displays that differ fundamentally from the stereotyped action pattern that defines yawning across vertebrate classes. Yawning is also contagious in a handful of social species, and some researchers have speculated that this could serve some derived communicative role, but these are not socially directed. In addition, the motor action pattern of contagious and spontaneous yawns is indistinguishable. While we acknowledge there might be variation in the duration of yawning that is correlated with a particular context or trigger (an empirical question), the physiological consequences resulting from the motor action pattern should still facilitate brain cooling via altering the rate and temperature of arterial blood traveling to the brain. Yawns that occur as a result of stress or anxiety, for example, are readily interpreted from the brain-cooling hypothesis (Miller et al., 2010 *Animal Behaviour*; Eldakar et al., 2017 *Adaptive Human Behavior and Physiology*), though it is unknown whether these yawns vary in duration.

Nevertheless, we were aware of the potential interference of such potentially socially triggered yawns, and removed all yawns from which we suspected that a conspecific or otherwise threatening situation may have triggered the yawn, as to provide a 'cleaner' test of our main hypothesis. We now explicitly mention this data-restriction measure in the methods section (see ll. 322-323).

2. The authors raise the issue of multicollinearity and suggest that this is a strong rationale for not examining relative brain size in their study. Multicollinearity can be an issue in studies of relative brain size and this is true of any variable that exhibits allometry. As far as I am aware, there is still no effective means of addressing this in a comparative context, but I do not agree that this means that it should not be tested at all. Further, by not including body size in all of their analyses, the authors are potentially ignoring an allometric effect on yawn duration. For example, all of the graphs in Figure 3 look like exponential (i.e., allometric) relationships that could be driven by body size and not brain size or neuron numbers. The brain:body ratio as a means of controlling body size is also not an effective means of incorporating overall size; ratios are often correlated with their denominators in evolutionary allometry. Thus, the brain:body ratio is probably correlated with body size, potentially indicating that body size is indeed exerting an effect on yawn duration. I would therefore suggest the following analyses: 1) examine body size explicitly in all

comparisons to determine if similar relationships occur across all variables; and 2) include body mass in models of brain size, neuron number, etc. and acknowledge that collinearity can affect the statistical modeling.

We agree with the reviewer that the found effects may (partly) be due to allometry, and therefore we initially included the brain:body ratio. Nevertheless, we acknowledge that this measure may not be ideal. Therefore, to address this and also the second reviewer's concern regarding allometry we have performed the following:

1. We ran phylogenetic correlations between all of our independent variables, including body weight, we find strong correlations of our brain measures with body size, and we now report this in tables S1 & S2). In addition, we have assessed the risk of multicollinearity for each model; i.e. bias in multivariate models including brain and body size measures as predictors of yawn duration in each model, using variance inflation factors (VIFs) (Zuur, Ieno, & Elphick, 2010).

If that risk was too high; i.e. a VIF > 3, it is suggested that phylogenetically controlled residuals should instead be utilized for identifying unique evolutionary relationships with relative brain size (Gutierrez-Ibanez, Iwaniuk, & Wylie, 2016). In those instances, we thus:

2a. calculated residual brain measures from a phylogenetically controlled linear regression of the respective brain measure on body size (Revell, 2009) and used these residuals as predictors in subsequent analyses. This was the case for both the mammalian and avian total brain size, and for mammalian total neuron count.

If, however, that risk was acceptably low; i.e. a VIF < 3, we instead

2b. included body size as an additional predictor in a multivariate regression analysis. This was the case for mammalian cortical neuron number, and avian total and pallial neuron numbers. **Note that** with regard to the latter two, based on the relative lack of raw correlations between these measures and body size, we did already include body size in our original models (/submission) regarding these avian brain measures, and independent of that body size found the reported associations between yawn duration and these avian brain measures.

Zuur, A. F., Ieno, E. N., & Elphick, C. S. (2010). A protocol for data exploration to avoid common statistical problems. *Methods in ecology and evolution*, 1, 3-14.

Gutierrez-Ibanez, C., Iwaniuk, A. N., & Wylie, D. R. (2016). Relative brain size is not correlated with display complexity in manakins: a reanalysis of Lindsay et al. (2015). *Brain, Behavior and Evolution*, 87, 223-226.

Revell, L. J. (2009). Size-correction and principal components for interspecific comparative studies. *Evolution*, 63, 3258-3268.

We now mention this in the methods section (ll. 424 ff) as follows:

“

It is common to adjust for body size in comparative studies of brain evolution by including body weight as an additional predictor in a multivariate analysis. However, recent work strongly suggests against this practice due to the high risk of inferential bias from multicollinearity (Rogell, Dowling, & Husby, 2019). When the risk of multicollinearity bias is high, it is suggested that phylogenetically controlled

residuals should instead be utilized for identifying unique evolutionary relationships with relative brain size (Gutierrez-Ibanez, Iwaniuk, & Wylie, 2016). To address this concern, we first used variance inflation factors (VIFs), which provide a standardized metric of multicollinearity, to quantify the risk of bias in multivariate models including brain and body size measures as predictors of yawn duration. VIFs of 1 indicate no multicollinearity, while $VIF > 3$ have been suggested to indicate undesirably high risk of multicollinearity bias in ecological datasets (Zuur, Ieno, & Elphick, 2010).

As expected given the raw and phylogenetic correlations (see Table S1-2), VIFs were undesirably high for the mammalian total brain size ($VIF = 13.05$) and total neuron count measures ($VIF = 5.10$), as well as for the avian total brain size measure ($VIF = 3.59$). In contrast, VIFs were sufficiently low for the mammalian total cortical neuron count ($VIF = 2.91$), as well as for the avian total neuron count ($VIF = 1.29$) and total pallial neuron count ($VIF = 1.06$). To ensure reliable inference for measures with $VIF > 3$ (i.e. mammalian total brain size and total neuron count, avian total brain size), we therefore calculated residual brain measures from a phylogenetically controlled linear regression of the respective brain measure on body size (Revell, 2009) and used these residuals as predictors in subsequent analyses. For measures with $VIF < 3$ (i.e. mammalian total neuron count, avian total neuron count and pallium neuron count), we instead included body size as an additional predictor in a multivariate regression analysis."

The results of these analyses are similar to the original analyses, although the effects of mammalian neuron numbers, both in total and in the cortex were not so strong anymore. We now report these results (ll. 149 ff) as follows:

"The aim of this study was to examine how absolute brain mass and neuron numbers drive differences in yawn duration across mammals and birds, yet models that do not control for body mass fail to rule out byproduct explanations due to allometric brain/body scaling. To avoid biases due to multicollinearity between body mass and the respective brain measures, we calculated residual brain mass measures from a phylogenetically controlled linear regression of body mass on the respective brain measure³⁴ and used these residuals as predictors in subsequent analyses. If, however, the risk of such a bias due to multicollinearity was acceptably low, we rather included body size as an additional predictor in the original models (see Methods for specifics).

Similar to the models of absolute brain size, model comparison did not support the inclusion of random slopes across either mammalian ($\Delta WAIC = -0.51 [1.90]$) or avian orders ($\Delta WAIC = -3.16 [3.12]$), suggesting that the observed effects of the body size adjusted brain measures on yawn duration were also consistent across taxa. Within mammals, after controlling for body size, clear positive associations remained between yawn duration and mammalian brain size ($\beta = 0.11 [0.07]$, 90% CI [0.00, 0.24], $p_+ = 0.95$, $d = 0.45 [0.28]$), with similarly sized but more statistically uncertain associations observed between yawn duration and the total mammalian neuron count ($\beta = 0.12 [0.14]$, 90% CI [-0.11, 0.35], $p_+ = 0.80$, $d = 0.42 [0.49]$) and cortical neuron count ($\beta = 0.13 [0.16]$, 90% CI [-0.16, 0.39], $p_+ = 0.78$, $d = 0.46 [0.58]$). The near equivalence of these effect sizes is expected, as the mammalian neuron counts exhibited very high phylogenetic correlations with total brain size (total neuron count: $r_{\text{phylo}} = 0.98$; cortical neuron count: $r_{\text{phylo}} = 0.93$), suggesting that these measures are providing similar information about brain evolution. Moreover, given the much smaller sample size for the neuron count measures ($N_{\text{species}} = 18-19$) as compared to total brain size ($N_{\text{species}} = 55$), which resulted in very low power for detecting small effect sizes (power = 0.08 for $\beta = 0.10$; see supplementary material), it is expected that the neuron count effects would nonetheless exhibit greater statistical uncertainty than total brain size.

Among avian taxa, after adjusting for body size, moderately sized positive associations remained between yawn duration and brain size ($\beta = 0.17 [0.10]$, 90% CI [0.00, 0.33], $p_+ = 1.00$, $d = 0.58 [0.35]$), as well as between yawn duration and total neuron count ($\beta = 0.20 [0.08]$, 90% CI [0.06, 0.34], $p_+ = 0.99$, $d = 0.67 [0.28]$) and pallium neuron count ($\beta = 0.18 [0.08]$, 90% CI [0.05, 0.30], $p_+ = 0.99$, $d = 0.61 [0.26]$). As in mammals, the phylogenetic correlations between the neuronal count and brain size

measures were very high across birds (total neuron count: $r_{\text{phylo}} = 0.94$; pallium neuron count: $r_{\text{phylo}} = 0.91$).”

It should, however, be noted that our primary hypotheses concern the evolution of yawn duration in response to brain size and neuron numbers, irrespective of whether these neurological measures evolve through direct selection or indirectly through selection on body size. Given these considerations, we therefore still conducted our primary statistical analyses without including body weight as an additional covariate, and added these new analyses after that.

While controlling for body size is crucial to ensure that the observed effects of brain size on yawn duration are not solely attributable to allometric scaling, this procedure also significantly reduces the variation in absolute brain measures which we hypothesized to predict yawn duration. In other words, by investigating brain size variation independent of body size, we are in essence testing a distinct biological hypothesis—i.e. whether brain size *deviations* from phylogenetically expected body size predict yawn duration—rather than our primary hypothesis about the physiological consequences of *absolute* brain size (irrespective of whether this brain size is expected given body size). Nonetheless, we still find clear effects of body size adjusted brain size on yawn duration in both birds and mammals, as well as for body size adjusted neural counts (both total and pallial) in birds, demonstrating the robustness of our findings. The increased statistical uncertainty observed for the nearly equivalent effects of body size adjusted mammalian neuron counts (both total and cortical) can also be clearly attributed to the loss of variability and statistical power caused by partialling out body size in the relatively small samples available for those measures ($n = 18$ & 19 respectively).

We discuss this in similar wording in the revised discussion of our manuscript (II. 233 ff)

3. In addition to the analyses I suggest above, the authors should examine neuronal density. Neuronal density is inversely correlated with brain size (and brain region size) and could provide further insights into the relationship between the brain and yawn duration.

While interesting, we had no a priori predictions for such analyses, and could also not think of a biological rationale other than to use density measures as a proxy for the complexity of the brain. Although we question how that adds to analyses about neuron numbers as these are a more direct way of testing (executive) functionality of the brain and subsequent need for cooling.

Nevertheless, to accommodate the reviewers curiosity, we have ran these analyses (see below).

We would like to first, however, point the reviewer to the fact that while the inverse relationship between brain size and neural density holds for many species, there is no correlation between density and brain size in primate brains, in parrot and songbird forebrains, and in anseriform brains, which may explain how the results of our analyses may deviate from the reviewers suggestion:

We find:

Cortical and pallium neural densities were negatively associated with body size in both mammals ($r = -0.89$, VIF = 4.13) and birds ($r = -0.85$, VIF = 4.31). Total neuronal density was also negatively correlated with body size, albeit with a much smaller effect size in mammals ($r = -0.09$, VIF = 1.07) as compared to birds ($r = -0.90$, VIF = 3.08). Consistent with this pattern, the relationship between total neuronal density and pallium density was much stronger in birds ($r = 0.98$) than the relationship between total neuronal density and cortical density in mammals ($r = 0.13$).

Among mammals, yawn duration was not clearly associated with total neuronal density ($\beta = -0.07$ [0.11], 90% CI [-0.26, 0.12], $p_+ = 0.27$, $d = -0.24$ [0.39]), nor with body size adjusted neuronal density ($\beta = -0.05$ [0.09], 90% CI [-0.20, 0.09], $p_+ = 0.39$, $d = -0.18$ [0.31]). Yawn duration exhibited a negative association with cortical density ($\beta = -0.28$ [0.10], 90% CI [-0.45, -0.10], $p_+ = 0.01$, $d = -1.01$ [0.36]), but this association disappeared once body size was adjusted for ($\beta = -0.03$ [0.13], 90% CI [-0.25, 0.19], $p_+ = 0.67$, $d = -0.11$ [0.46]).

Yawn duration also had a moderately sized association with total neural density across birds ($\beta = -0.15$ [0.10], 90% CI [-0.31, 0.02], $p_+ = 0.07$, $d = -0.49$ [0.33]), but no association was found after adjusting for body size ($\beta = 0.01$ [0.09], 90% CI [-0.14, 0.15], $p_+ = 0.52$, $d = 0.02$ [0.30]). Pallium neuron density also did not clearly associate with yawn duration ($\beta = -0.09$ [0.11], 90% CI [-0.28, 0.08], $p_+ = 0.20$, $d = -0.31$ [0.36]), nor when adjusting for body size ($\beta = 0.05$ [0.09], 90% CI [-0.11, 0.21], $p_+ = 0.70$, $d = 0.17$ [0.32]). Taken together, these results suggest that there is no unique relationship between yawn duration and either total or cortical neuron density, with the observed associations for unadjusted measures being entirely accounted for by variation in body size.

As we find no clear patterns regarding a potential link between yawn duration and neural density, and as mentioned, we have no biologically relevant, a priori predictions about this measure, we have decided to leave these analyses out of the manuscript, but would of course, given a plausible hypothesis be willing to put them in the supplemental materials.

4. The Discussion places a lot of emphasis on the brain cooling hypothesis, but it remained a bit unclear to me why a correlation between brain size and yawn duration would be necessary. If the primary function of yawning is to cool the brain, does this imply that larger brains have greater heat dissipation needs? Or is it that larger bodies generate and retain more heat and therefore might require longer yawns in order to cool the larger brains present in larger animals? Including the analyses I suggested would address these questions.

Larger brains have greater heat dissipation needs. We have now tried to further clarify the rationale for this prediction in the paper (ll. 68 ff & 217 ff).

5. Is it possible that the extended yawns in the naked mole-rat are unrelated to thermoregulation of the brain and reflect some other function of yawning? It is a highly social species, so I was thinking about the possible use of yawns as a social signal or to reflect stress/anxiety.

We now discuss the possibility that yawns serve a social function in this species, and call for future research in this area (see ll. 296-297).

6. I am not sure that I agree with the authors that the data supports a brain cooling function based on what was presented. There is little doubt that brain size and neuron numbers have evolved in concert with longer yawns, but without controlling for body size, this could reflect allometry or something else. I think it certainly points to an interesting example of correlated evolution, but the link to the brain cooling theory needs to be explained in a more explicit fashion.

We now correct for body size (see our detailed response to your second remark), and therefore can rule out allometry. We also explicitly describe why brain size should coevolve with yawn duration based on the brain cooling hypothesis and have revised the final sentence of the Discussion to indicate that the data support the distinct predictions of this hypothesis.

Reviewer #2 (Remarks to the Author):

In this paper the authors perform a comparative study aimed at testing the 'brain cooling hypothesis' for yawning, by examining correlations between brain size, other neural traits and yawn duration. I was interested to read the paper having read previous comparative analyses from some of the same authors. I didn't find these previous analyses convincing due to the modest sample sizes (in most cases) and some statistical issues, in particular the lack of phylogenetic correction (ref 15), use of multiple correlated traits and ratios (refs 15, 16), and the use of brain size estimates derived from transformations of body size (ref 17). Indeed, the authors acknowledge some of these issues on lines 75-77, which is good to see. In any case, it was interesting to see the topic revisited with larger samples and potentially more robust methods.

Unfortunately, although the paper is nicely written and the Bayesian analyses seem to be well implemented, I still think some of these basic concerns remain. In particular:

1. No evidence is presented that the associations are specific to brain mass, or any other neural trait – I'd repeat the analyses with body mass to compare fit. If you can't find evidence brain mass predicts the yawn data better than body mass I don't think your conclusions are justified.

As suggested by the reviewer, we have now rerun our analyses including either body mass or on the residuals of the respective brain measures on body mass. Please see our elaborate explanation in response to the 2nd comment of reviewer 1.

2. Ratios are not a valid measure of relative brain size as brain and body mass does not scale isometrically.

As we have now used an alternative method to control for body size (see above), we have decided to omit all analyses on brain-to-body mass ratios.

3. The results for brain mass, neuron number, cortical neuron number are treated as if they are independent of each other, but they are not. As such, they don't provide independent support for the hypothesis.

Please see our detailed response with regard to this issue below.

As an aside, none of the dataverse links worked for me (I get a 'page not found' error) so I have been unable to look at these in detail.

We apologize that the links do not work when clicking on them. However, if you copy-paste the links into your browser it should work.

Specific points in more detail:

- The collection of the yawn data is well described and seems as robust as it can be. I'd like to see more detail on how species identity was made for YouTube (and other public) videos. I'd also point

out that many YouTube links are not stable, for example in some of the previous papers the links no longer work. But I'm not sure how this can be accounted for, unless the authors download all the videos, but that seems a bit arduous...

Species identity was checked by two of the authors and checked again if any doubt still existed by the senior author on the paper. We now explicitly mention this in the methods section (l. 323)

As for the YouTube videos, in fact we have downloaded the videos from YouTube, yet because we don't hold the copyrights to these videos, we cannot add these to our dataverse, although we can make these available upon request.

- I think it is justified to analyse brain mass without including body weight in the analysis if your hypothesis concerns absolute size, provided the same analyses are also performed with body mass allowing direct comparisons of model fit. If the brain models fit better than the body models you can claim a main effect of brain mass, if not, you can't rule out general associations with body mass and I don't think your inferences would be supported in that case. The neuron number data may be your best bet here, since neuron number varies with body mass less consistently than brain mass. If you can't test for independent effects of neuron number and body mass then I would be pretty unconvinced.

This indeed is our hypothesis, but we have now also gone on to show that brain mass still predicts yawn duration even after correcting for body size (see above). Thus, we can rule out allometric scaling.

- In addition, if you're justifying not including body mass in the analyses due to the effects of collinearity you should probably test that this is actually happening in your data. Alternatively, you can potentially try to side step the problem to some extent by regressing both yawn duration and brain mass against body mass and then calculate and compare the residual values to check that variation in these traits that is independent of body size are correlated, as your argument would predict. This is not ideal as residuals are not data, but it would provide another way to bolster your conclusion and might be a useful supplementary analysis if collinearity is a problem.

We would like to thank the reviewer for their suggestion regarding calculating the residuals, as this is now the approach we took (see above). As for the problem of collinearity, in our original submission we already reported raw correlations between our log-transformed brain measures and the log-transformed body size measure with an r of 0.9 or higher, which clearly suggests that we would run into problems with regard to multicollinearity. Nevertheless, in our revised paper we now also explicitly checked for the risk of multicollinearity bias using variance inflation factors (VIFs). Please see our response to the 2nd comment of reviewer 1 for more details.

- I don't agree brain/body ratios are statistically valid measures of relative brain size as they do not appropriately correct for brain~body allometry. These analyses should be replaced with multiple regressions (e.g. a basic model of yawn ~ body mass + brain mass). I don't believe any analysis based on ratios is robust when the traits involve do not scale isometrically. The ratios will probably still correlate with body size.

We have now included the suggested analyses which control for allometric scaling rules (see above) and have removed the original analyses on brain/body ratios.

- Including multiple neural traits in the analyses is complicated by the fact that they are all correlated with each other, and therefore not statistically independent. In this context its interesting the beta estimates tend to be very similar across traits within a dataset. You don't really compare the results with brain mass or neuron number, but they are presented as if they provide independent lines of support for your hypothesis. Instead, you could use these data to confirm the most meaningful trait, either by comparing fit of models with different traits, for example. When comparing models obtained with different traits they should be based on the same sampling of species, so you would need to subsample the brain dataset and repeat the analysis to compare it to the neuron number dataset, for example. Regression models can also be built that narrow down the effect, e.g. is the effect due to neuron number, independent of the effects of brain size, or vice versa which might suggest non-neuronal cells are important.

Whereas we agree that these measures are clearly not independent from each other, we do feel that we should report on them independently, as we have clear a priori predictions that the duration of yawning (as a cooling mechanism) should increase with brain size (brain mass) and brain complexity (i.e. neuron number and cortical/pallial neuron number as proxies for (executive) functioning / brain complexity).

Nonetheless, we agree that it would be misleading to not emphasize to the reader that these measures are providing largely equivalent information in our analysis, due to the very high phylogenetic correlations observed between these measures. We have therefore added the following text to our results:

"The near equivalence of these effect sizes is unsurprising, as the mammalian neuron counts exhibited very high phylogenetic correlations with total brain size (total neuron count: $r_{phylo} = 0.98$; cortical neuron count: $r_{phylo} = 0.93$), suggesting that these measures are providing nearly equivalent information about brain evolution ... As in mammals, the phylogenetic correlations between the neuronal count and brain size measures were very high across birds (total neuron count: $r_{phylo} = 0.94$; pallium neuron count: $r_{phylo} = 0.91$)". (ll. 168-183)

We appreciate the reviewer's additional suggestion to perform model comparison to determine which of the three brain measures is most important for predicting yawn duration. We would utilize this analytic strategy if there had been more independent variation observed in our dataset. However, given the very high phylogenetic correlations (all >0.9 ; see new Table S1-2) and the approximately equivalent body size adjusted effect sizes (mammals β 's = 0.12-0.13; avian β 's = 0.17-0.20), we can be confident that little will be gained by attempting to further partial out meaningful biological distinctions from the highly correlated effects of these measures.

To encourage further attention to this issue in future research, we have also added the following text to the discussion:

"Indeed, phylogenetic correlations were very high between all brain measures (see Table S1-2), suggesting that the total brain size and neuron count measures are providing nearly equivalent information about the evolution of yawn duration in our analyses. Future studies would benefit from investigating these associations within lineages where neuronal count and total brain size exhibit more independent variation than observed in the present study, which would facilitate further teasing apart the distinct causal effects of brain size and brain complexity on yawn duration." (ll. 252 ff)

- Another trait I would like to see included in the analyses is lung mass or volume, as there (to me) it

seems reasonable to expect that yawn duration is constrained (or correlated) with lung capacity. There is data available for mammals at least, in Navarrete, A., van Schaik, C. P., & Isler, K. (2011). Energetics and the evolution of human brain size. *Nature*, 480(7375), 91-93.

This comment seems to be based on the long-lasting idea that yawning somehow aids in oxygenation, yet, as we already mention in our introduction, previous research has already refuted this idea. Alternatively, the reviewer may have hinted at the brain-cooling function of inhaling cool air while yawning and that this may be constrained by lung capacity, but here we should note that although this is part of the mechanism, the gaping of the jaw, and the subsequent increased blood flow to the brain, appears to be the most important feature of yawning (as opposed to normal inhalation) with regard to the brain-cooling hypothesis. Thus, we do not believe there is a strong reason to expect yawning to be constrained by lung capacity.

Nevertheless, to accommodate the reviewer's curiosity, we have ran said analyses. Note, however, that lung mass data from Navarrete et al. (2011) was only available for mammalian species, and of the sample of which we have yawn data only 10:

Within the small sample of mammals for which accurate data were available ($N_{\text{species}} = 10$), we found that yawn duration exhibited a strong positive association with overall lung capacity ($\beta = 0.40$ [0.07], 90% CI [0.27, 0.51], $p_+ = 1.00$, $d = 1.56$ [0.28]). However, this association entirely disappeared after adjusting for body size ($\beta = 0.02$ [0.11], 90% CI [-0.19, 0.21], $p_+ = 0.58$, $d = 0.09$ [0.43]). Given that lung capacity is nearly perfectly correlated with body size in this reduced sample ($r = 0.99$, VIF = 32.59), the unadjusted association between yawn duration and lung capacity appears to be a spurious correlation arising from common allometric scaling with body size, rather than any direct causal relationship.

Given these inconclusive results (also given the small sample), and the lack of any a priori hypothesis, we decided not to include this analysis in our manuscript.

- Can the authors report the strength of phylogenetic signal, and how this was incorporated in the analyses? Do the models assuming high/complete phylogenetic signal? Or is it estimated at the same time as the other parameters.

Our Bayesian models estimated the expected phylogenetic signal along with the other model parameters during model fitting, rather than introducing a priori assumptions about signal strength which may have biased inference. We have now added text describing the estimated phylogenetic signals for yawn duration, prior to partialling out the various brain measures, in mammals and birds:

"Within this final sample, moderate to large phylogenetic signals were estimated for yawn duration in mammals ($\lambda = 0.80$) and birds ($\lambda = 0.64$)." (ll. 456-457).

- Line 529-530: can you show that it does so, rather than just saying it does? Similarly, how was the gamma distribution detected?

A Gamma distribution was utilized for our phylogenetic models based on a priori statistical considerations, as this distribution relaxes the assumption of constant variance (i.e. homoscedasticity) in linear phylogenetic models, which is routinely violated with non-negative, right skewed data such as our yawn duration measures. For this reason, Gamma distributions are increasingly suggested for investigating duration measures such as response latencies, time-to-event data, etc.

E.g.

Lo, S., & Andrews, S. (2015). To transform or not to transform: Using generalized linear mixed models to analyse reaction time data. *Frontiers in Psychology*, 6, 1171.

Ng, V. K., & Cribbie, R. A. (2017). Using the gamma generalized linear model for modeling continuous, skewed and heteroscedastic outcomes in psychology. *Current Psychology*, 36(2), 225-235.

The rate and dispersion parameters of these Gamma models are estimated during the model fitting procedure. To ensure that these models appropriately accounted for the distribution of our observed data, we utilized a Bayesian diagnostic procedure called posterior predictive checks. Put simply, this involves predicting the expected values for each data point based on observed covariate values and comparing the distribution of predicted values to the raw data (indicated by the black line in the figures below). Because Bayesian analyses appropriately account for the effects of parameter uncertainty on predictions, multiple independent predictions are generated to compare absolute fit to the data, as indicated by the multiple light blue lines below representing predictions from 100 random posterior MCMC draws. As is indicated by the substantial overlap between observed and predicted values for both mammalian and avian yawn duration, our Gamma phylogenetic models are predicting the non-Gaussian distribution of our data quite well, consistent with a priori statistical expectations.

- Figure 2: I find these figures a little uninformative, if you replaces the plots of brain mass with plots of body mass they would look almost identical.

-Given our hypothesis that brain size should drive differences in yawn duration, we find this figure to be worthwhile and informative, especially given our new analyses that correct for allometric scaling rules.

In sum, I think this is an interesting topic, and the authors explain the logic behind the analyses very well. I just don't think they explore the data sufficiently to make the argument that these associations have anything to do with brain size specifically. As it currently stands my interpretation of the results

would simply be that bigger birds/mammals have longer yawns. However, I do think there are ways of analysing and supplementing these data to make the authors' argument more robust and I hope the suggestions above help.

We have addressed the major concern regarding allometry. We would like to thank the reviewer for the highly constructive feedback on this paper, and indeed feel that our argument is more robust with these additional analyses.

Reviewers' comments:

Reviewer #1 (Remarks to the Author):

The authors have responded to all of my major concerns and the revised analysis that includes body mass does support their original analyses. The manuscript is well written and presented and I have only three minor concerns/questions about the updated manuscript that I outline below.

1. In Figure 3, the x-axis of all scatterplots is "standardized", but when I looked for how this was done in the methods (lines 485-487), the information provided was a bit vague. Can the authors please explain specifically how this was done? Also, it appears from this section that the standardization was done to make comparisons across measurements, but I am not understanding why this is necessary.

2. The inclusion of body size measurements in the models greatly improved the strength of the overall study, but the scatterplots for these analyses need to be presented to assess how the absolute measures compare with the relative measures. Please add these additional plots to Figure 3 or add a Figure 4.

3. With respect to neuronal density, the authors and I will have to agree to disagree. In my opinion, neuronal density could be an important metric in relation to brain cooling. A brain that has a higher neuronal density could generate more heat per unit volume and therefore be in greater need of cooling, therefore leading to longer yawn duration. Lack of a significant effect might not, however, be that meaningful in the current analysis given the much smaller sample size and variation in the relationship between neuronal density and brain size across taxa. So, I think including in the supplemental material would prove useful to other researchers, but it does not need a lot space dedicated to those analyses in the manuscript proper.

Reviewer #2 (Remarks to the Author):

The authors have taken on board my previous comments and tried to address them, and I do appreciate them doing so. However, I still have some concerns about some of the analyses:

1. Use of residuals

The authors have included tests of the association between yawn duration and residual brain size (i.e. residual variance around a brain size \sim body size). This is OK but not quite what I previously suggested which was "regressing both yawn duration and brain mass against body mass and then calculate and compare the residual values to check that variation in these traits that is independent of body size are correlated". As the authors themselves note, using residual brain size with yawn duration alters the test in a way that doesn't fit with their biological hypothesis. What I was suggesting was to calculate residuals from BOTH a brain \sim body regression and a yawn \sim body regression. If the effects of the yawn \sim brain association are causative, then variation in both traits that are independent of body size, should still be correlated. In my mind, this fits closer to their biological argument and I would hold off judging the robustness of the results until seeing those results.

As it is, the effects with relative brain size are not particularly strong, either by $P+$ (if you interpret this in an analogous way to frequentist p-values, which is admittedly probably philosophically wrong!) or the effect size (even before being pedantic about multiple testing). It's not entirely clear from the way the results are written which results involve multiple regressions with brain and body size and which involve residual brain size, but as I understand it, the results in the multiple regressions also seem weaker, which is perhaps concerning since this is a more 'proper' statistical test. But I imagine the comparisons here are potentially conflated by sample size.

There are also no comparisons of model fit, either between neural comparisons or between brain and body size (see below as well).

2. Lung volume

This idea was not based on any particular physiological hypothesis other than the assumption large lungs take longer to fill, so yawn duration probably scales with lung volume. I still think this is a valid point and, as a priori hypotheses go, pretty intuitive. I appreciate the authors trying this but the low sample size conflates their interpretations here – if you repeat the brain mass/relative brain size comparisons with the sample sample as available for lung mass, does that apparent effect also go away? If they want to reject this properly the authors would need to increase the sample size here.

Elsewhere in their response the authors write: “It should, however, be noted that our primary hypotheses concern the evolution of yawn duration in response to brain size and neuron numbers, irrespective of whether these neurological measures evolve through direct selection or indirectly through selection on body size.” I feel this is problematic thinking, their biological hypothesis is that there is a causative link between neuron number and yawn duration – so this should be the observed effect. To test this hypothesis robustly, the authors need to convincingly argue against an effect of body size or other traits linked to body size.

3. Neuron densities

I agree with the authors that analyses of neuron density (suggested by reviewer one) aren't necessary to include. But, one thing I perhaps missed in the last round (or may be new, not sure) is the comparison between mammals and birds which shows mammals have longer yawn durations than predicted would be predicted by the yawn \sim brain/body relationship in mammals. I'm curious how the authors interpret this in the context that their hypothesis is about heat produced by neuronal cells, and therefore about neuron number. Given (some) birds have higher densities than (some) mammals, would you not predict the opposite pattern? i.e. a longer yawn durations in birds than mammals, for a given brain size.

Similarly, by this logic I'd expect the fit of the model with neuron number to be higher than the fit with brain mass (when based on the same sample set), especially because we know the brain \sim neuron number relationship varies across taxa so there should be reasonable power to test this. But I see no test of this, instead the authors say they are “confident” they would see no effect due to collinearity – this is not an argument against doing this. The comparisons should be made, presented and discussed regardless of the results, otherwise it is impossible for the reader to properly evaluate their robustness. This could include estimates of variance inflation, if the authors feel this is necessary.

In sum, although the authors have made some attempts to address my concerns that the associations between brain and yawn duration are indirect, or not robust. However, in some cases the authors have argued against what I feel are critical tests (model fit comparisons), have performed tests in an incomplete manner that favours their current interpretation (the lung comparison) or have not quite followed what I was suggesting previously (the residuals analysis). Unfortunately therefore, my concerns remain and I'm yet to be convinced their conclusions are correct. Of course, the authors have the right to disagree, and I hope they do not think I am being stubborn or unreasonable! Sorry to be the Christmas Grinch, I do find the paper and hypotheses interesting, but the field is full of spurious correlations between brain size and various traits, so I think it is important we fully dissect the data before making big claims.

Typos:

Abstract: line 27: perhaps say “multiple, non-independent neural traits including...”

Line 38: comparative analyses showing what?

Line 59: "following" > followed

Line 150: seems odd to have this as separate section, its necessary to test the primary hypothesis

Reviewers' comments:

Reviewer #1 (Remarks to the Author):

The authors have responded to all of my major concerns and the revised analysis that includes body mass does support their original analyses. The manuscript is well written and presented and I have only three minor concerns/questions about the updated manuscript that I outline below.

1. In Figure 3, the x-axis of all scatterplots is “standardized”, but when I looked for how this was done in the methods (lines 485-487), the information provided was a bit vague. Can the authors please explain specifically how this was done? Also, it appears from this section that the standardization was done to make comparisons across measurements, but I am not understanding why this is necessary.

Thank you very much for pointing this out. We acknowledge that the information we provided here was a bit poor. Values were standardized to z-scores (i.e. $[\text{value} - \text{mean}] / \text{SD}$) to facilitate effect size comparison across brain measures. We now mention this in the text (l. 537). Note that this does not in any way change the distribution of the data but simply re-expresses the scale of the values to derive standardized regression coefficients. This allowed us to compare effect sizes of the measures with differing degrees of variance within and across mammals and birds on a comparable scale.

2. The inclusion of body size measurements in the models greatly improved the strength of the overall study, but the scatterplots for these analyses need to be presented to assess how the absolute measures compare with the relative measures. Please add these additional plots to Figure 3 or add a Figure 4.

We have added a Figure 4 as suggested.

3. With respect to neuronal density, the authors and I will have to agree to disagree. In my opinion, neuronal density could be an important metric in relation to brain cooling. A brain that has a higher neuronal density could generate more heat per unit volume and therefore be in greater need of cooling, therefore leading to longer yawn duration. Lack of a significant effect might not, however, be that meaningful in the current analysis given the much smaller sample size and variation in the relationship between neuronal density and brain size across taxa. So, I think including in the supplemental material would prove useful to other researchers, but it does not need a lot space dedicated to those analyses in the manuscript proper.

We have now followed the reviewer’s suggestion and mention these analyses in the main text, while referring the reader to the supplements for further detail (l.225ff).

Reviewer #2 (Remarks to the Author):

The authors have taken on board my previous comments and tried to address them, and I do appreciate them doing so. However, I still have some concerns about some of the analyses:

1. Use of residuals

The authors have included tests of the association between yawn duration and residual brain size (i.e. residual variance around a brain size ~ body size). This is OK but not quite what I previously suggested which was “regressing both yawn duration and brain mass against body mass and then calculate and compare the residual values to check that variation in these traits that is independent of body size are correlated”. As the authors themselves not, using residual brain size with yawn duration alters the test in a way that doesn’t fit with their biological hypothesis. What I was suggesting was to calculate residuals from BOTH a brain~body regression and a yawn~body regression. If the effects of the yawn~brain association are causative, then variation in both traits that are independent of body size, should still be correlated. In my mind, this fits closer to their biological argument and I would hold off judging the robustness of the results until seeing those results.

As it is, the effects with relative brain size are not particularly strong, either by P+ (if you interpret this in an analogous way to frequentist p-values, which is admittedly probably philosophically wrong!) or the effect size (even before being pedantic about multiple testing). It’s not entirely clear from the way the results are written which results involve multiple regressions with brain and body size and which involve residual brain size, but as I understand it, the results in the multiple regressions also seem weaker, which is perhaps concerning since this is a more ‘proper’ statistical test. But I imagine the comparisons here are potentially conflated by sample size.

We disagree with the reviewer about the approach to be taken here. To be specific, the alternative test suggested in their comment is unnecessary. Taking the residuals of brain size on body size, as we did, necessarily means that any association between brain size residuals and yawn duration is **independent** of the association between brain size and body size. And that is exactly what we want to show, as it demonstrates that, independent of changes in body size, increases in brain size are associated with increases in yawn duration. There is no need to additionally take the residuals of yawn duration on body size before conducting this analysis. This is not standard practice in phylogenetic comparative research, nor has it been recommended in any methods papers we are aware of.

However, to further examine the robustness of these findings, we now followed the suggestion of this reviewer and also conducted additional analyses in which we first used Gaussian phylogenetic regressions to partial out the effects of body size on *both* brain and yawn duration prior to estimating their association. This approach relied on

the simplifying assumption that yawn duration residuals are normally distributed, and we therefore used classical linear PGLS models for assessing the association among these residual values.

Consistent with the findings using more robust Bayesian methods, significant associations between yawn duration and all brain measures were observed across mammals and birds, and we now mention this in the main text (I.209ff), while also referring to the supplementary materials for the details: i.e.,

“Consistent with the findings reported before using more robust Bayesian methods, significant associations between yawn duration and brain measures were observed across mammals (brain size: $t = 2.90$, $p = 0.005$; neuron count: $t = 2.56$, $p = 0.02$; cortical neuron count: $t = 2.18$, $p = 0.04$) and birds (brain size: $t = 2.57$, $p = 0.01$; neuron count: $t = 2.42$, $p = 0.02$; cortical neuron count: $t = 2.50$, $p = 0.02$) after partialling body size from both the brain and yawn duration data. It should be noted that these simpler analyses provide stronger support for the effects of neuron counts on yawn duration independent of body size, as compared to those reported in the main text using more conservative Bayesian priors and appropriate non-Gaussian distributions to account for heteroscedasticity in yawn duration.”

Therefore, it should be clear that we have not selectively reported our findings so as to artificially inflate the support provided for our hypotheses. Rather, we have reported more conservative estimates in the main text because we think they provide a more accurate representation of the evolutionary relationship we sought to investigate than the standard PGLS analyses reported here, which only further bolster our findings.

As for the comment about multiple testing: This is incorrect. As we explain in our methods (and support with appropriate references), we used regularizing priors in our Bayesian model that are specifically designed to produce conservative estimates and reduce inferential errors due to multiple testing and sampling error.

We were also surprised with the comment that it is unclear which tests were run on the residuals and which were not, as we dedicated two paragraphs to this issue, one for the analyses on the original values, one for the analyses on the residuals, and have explained that in detail in the main text and the methods.

There are also no comparisons of model fit, either between neural comparisons or between brain and body size (see below as well).

See our response below.

2. Lung volume

This idea was not based on any particular physiological hypothesis other than the assumption large lungs take longer to fill, so yawn duration probably scales with lung

volume. I still think this is a valid point and, as a priori hypotheses go, pretty intuitive. I appreciate the authors trying this but the low sample size is conflates their interpretations here – if you repeat the brain mass/relative brain size comparisons with the sample as available for lung mass, does that apparent effect also go away? If they want to reject this properly the authors would need to increase the sample size here.

We are afraid that we have to disagree with the reviewer again. First, we want to reiterate that we did not have any a priori hypothesis about lung volume, and in fact have argued why there should be no relationship. Nevertheless, to accommodate the specific request of this reviewer we have performed the analyses with the sample of lung volumes offered to us (i.e., the species that were both in the manuscript the reviewer referred us to, and in our sample). The results from this exploratory analysis were negative, which could be due to the low sample size. However, the fact that there is not a larger sample to draw from is not our responsibility, and the comment that we need to increase our sample on this regard seems, to be honest, a bit out of line. Similarly, the comment of this reviewer (further below) that “(we) have performed tests in an incomplete manner that favours their (*our*) current interpretation (regarding the lung comparison)”, is simply false and is quite offensive. As mentioned above, we have, as per request of this reviewer, performed the requested test with the available sample.

Elsewhere in their response the authors write: “It should, however, be noted that our primary hypotheses concern the evolution of yawn duration in response to brain size and neuron numbers, irrespective of whether these neurological measures evolve through direct selection or indirectly through selection on body size. “ I feel this is problematic thinking, their biological hypothesis is that there is a causative link between neuron number and yawn duration – so this should be the observed effect. To test this hypothesis robustly, the authors need to convincingly argue against an effect of body size or other traits linked to body size.

We agree with this reviewer that we need to actively argue against an effect of body size to argue against alternative hypotheses that are based on body size rather than brain size, and we do so now. However, we want to reiterate that our hypothesis is about the cooling effect of yawning on absolute brain size/heat produced by the whole brain, and not so much on the residuals. As such it really is not important what evolutionary pathway has led to the brain size and neuron numbers we observe and use in the analysis.

Current comparative literature is fraught with obsession to treat body size as a statistical nuisance. Authors often calculate relative or residual size of various traits and fail to realize that **absolute sizes have biological meaning**. For instance, production of heat and/or processing power of the brain will certainly correlate much better with absolute brain size or absolute number of neurons than with relative brain size or number of neurons per a given body size.

3. Neuron densities

I agree with the authors that analyses of neuron density (suggested by reviewer one)

aren't necessary to include. But, one thing I perhaps missed in the last round (or may be new, not sure) is the comparison between mammals and birds which shows mammals have longer yawn durations than predicted would be predicted by the yawn ~brain/body relationship in mammals. I'm curious how the authors interpret this in the context that their hypothesis is about heat produced by neuronal cells, and therefore about neuron number. Given (some) birds have higher densities than (some) mammals, would you not predict the opposite pattern? i.e. a longer yawn durations in birds than mammals, for a given brain size.

This is an interesting question, one that has also crossed our mind and we have dived a bit further into this. However, first we need to note that indeed it is the case that only **some** bird species show higher neuron densities than **some** mammals, notably birds belonging to the clade core landbirds (Telluraves). And while these lineages are represented in our data, so are more basal lineages that do not have such impressive neuronal densities.

Most importantly, however, avian neurons are on average much smaller and likely have much less extensive dendritic arbors than mammalian neurons. Thus, an average avian neuron requires much less energy to maintain membrane potential than an average mammalian neuron. Indeed, it has been recently shown that an average pigeon neuron is on average 3-fold energetically cheaper than an average mammalian neuron (von Eugen et al. 2018, Poster # 068.12/QQ22. 2018 Neuroscience Meeting Planner. San Diego, CA). These results strongly suggest that despite high neuronal densities avian brains may actually produce less heat than mammalian brains of equal size. We now discuss this in the revised version of our paper (I.335ff) next to the other potential reasons for the difference in yawn duration between birds and mammals that were already discussed in the previous version of the paper.

Similarly, by this logic I'd expect the fit of the model with neuron number to be higher than the fit with brain mass (when based on the same sample set), especially because we know the brain~neuron number relationship varies across taxa so there should be reasonable power to test this. But I see no test of this, instead the authors say they are "confident" they would see no effect due to collinearity – this is not an argument against doing this. The comparisons should be made, presented and discussed regardless of the results, otherwise it is impossible for the reader to properly evaluate their robustness. This could include estimates of variance inflation, if the authors feel this is necessary.

We have now included comparisons of fit for the models on the different brain measures. However, as predicted due to the smaller samples with regard to the neuron counts and the strong correlations between the different neurological measures, these comparisons did not render any clear differences between the models. We now mention this in the main text (I.218ff), while adding a cautionary note regarding the interpretation due to the small(er) sample size and the collinearity, and while referring to the supplementary materials for detail.

In sum, although the authors have made some attempts to address my concerns that the associations between brain and yawn duration are indirect, or not robust. However, in some cases the authors have argued against what I feel are critical tests (model fit comparisons), have performed tests in an incomplete manner that favours their current interpretation (the lung comparison) or have not quite followed what I was suggesting previously (the residuals analysis). Unfortunately therefore, my concerns remain and I'm yet to be convinced their conclusions are correct. Of course, the authors have the right to disagree, and I hope they do not think I am being stubborn or unreasonable! Sorry to be the Christmas Grinch, I do find the paper and hypotheses interesting, but the field is full of spurious correlations between brain size and various traits, so I think it is important we fully dissect the data before making big claims.

We understand the reviewer's concerns, yet do not agree with all their arguments.

Typos:

Abstract: line 27: perhaps say "multiple, non-independent neural traits including..."

Whereas we understand the reviewer's point here, we would like to keep the current wording as it is much more descriptive about what we exactly did, and we fully expect that the readership of Communications Biology understands that these traits are not independent.

Line 38: comparative analyses showing what?

The comparative analyses show that yawning is widespread across diverse species of vertebrates.

Line 59: "following" > followed

Changed as suggested

Line 150: seems odd to have this as separate section, its necessary to test the primary hypothesis

We agree that it is necessary to falsify alternative hypotheses based on body size, but not to test the primary hypothesis, which is about absolute brain size or neuron numbers and yawn duration (see our response above). Therefore, we remain of the opinion that this should be presented as a separate section.